# Meis1 establishes the pre-hemogenic endothelial state prior to Runx1 expression

Patrick Coulombe [1,2], Grace Cole[1,3], Amanda Fentiman[1,4], Jeremy D. K. Parker[1], Eric Yung [5], Misha Bilenky[1], Lemlem Degefie [1], Patrick Lac[1], Maggie Y. M. Ling [1], Derek Tam[1], R. Keith Humphries[5,6] & Aly Karsan [1,2,3,4] ✉

Hematopoietic stem and progenitor cells (HSPCs) originate from an endothelial-to-hematopoietic transition (EHT) during embryogenesis. Characterization of early hemogenic endothelial (HE) cells is required to understand what drives hemogenic specification and to accurately define cells capable of undergoing EHT. Using Cellular Indexing of Transcriptomes and Epitopes by Sequencing (CITE-seq), we define the early subpopulation of pre-HE cells based on both surface markers and transcriptomes. We identify the transcription factor Meis1 as an essential regulator of hemogenic cell specification in the embryo prior to *Runx1* expression. *Meis1* is expressed at the earliest stages of EHT and distinguishes pre-HE cells primed towards the hemogenic trajectory from the arterial endothelial cells that continue towards a vascular fate. Endothelial-specific deletion of *Meis1* impairs the formation of functional *Runx1*-expressing HE which significantly impedes the emergence of pre-HSPC via EHT. Our findings implicate *Meis1* in a critical fate-determining step for establishing EHT potential in endothelial cells.

Hematopoietic stem and progenitor cells (HSPCs) originate from an endothelial-to-hematopoietic transition (EHT) during embryogenesis. This is a conserved process amongst vertebrates, characterized by the emergence of intra-aortic hematopoietic clusters (IAHC) in the major arteries of the embryo[1,2]. In the mouse embryo, IAHC are detected around embryonic day (E)10.5 mainly in the dorsal aorta of the aorta-gonad-mesonephros (AGM) region. Expression of the surface marker CD117 (c-Kit) labels most IAHC cells from which maturing pre-HSPCs have been isolated using hematopoietic commitment markers such as CD41, CD43, and CD45[1,3–5]. Within the most mature CD45-expressing pre-HSPC, CD27 has been shown to further enrich for pre-HSPCs with long-term engraftment capabilities[6]. While the gradual onset of expression of these surface markers is widely used to segregate pre-HSPC at different stages of maturation along the EHT trajectory, cells with the ability to initiate EHT are harder to isolate, and far less is

known about what differentiates vascular endothelium from the rare endothelial cells (EC) that acquire hemogenic capacity[7,8].

It is estimated that only about 1.3% of EC isolated from E10.5 dorsal aorta have functional hematopoietic potential ex vivo[9,10]. These hemogenic endothelial (HE) cells are morphologically similar to vascular EC and express classical endothelial surface markers such as VE-Cadherin (Cdh5) and CD31 (Pecam1)[8]. Therefore, defining what initially distinguishes the rare HE cells capable of forming HSPC from the remaining vascular endothelium is challenging. In an attempt to prospectively enrich and study HE cells, several reporter models have been developed based on key EHT genes such as *Sca1* (Ly6a-GFP), *Runx1* (*Runx1*+23GFP), and *Gfi1* (*Gfi1*+/tomato)[9,11–13]. *Runx1*-deficient embryos are devoid of IAHCs[14] and the *Runx1*+23 conserved noncoding element (CNE) has been shown to be active in HE and all IAHC[15], making it a useful reporter to track cells after they have acquired

[1]Michael Smith Genome Sciences Centre, BC Cancer, 675 West 10th Avenue, Vancouver, BC V5Z 1L3, Canada. [2]Department of Experimental Medicine, University of British Columbia, Vancouver, BC V6T 2B5, Canada. [3]Department of Pathology and Laboratory Medicine, University of British Columbia, Vancouver, BC V6T 2B5, Canada. [4]Interdisciplinary Oncology Program, University of British Columbia, Vancouver, BC V6T 2B5, Canada. [5]Terry Fox Laboratory, BC Cancer, 675 West 10th Avenue, Vancouver, BC V5Z 1L3, Canada. [6]Department of Medical Genetics, University of British Columbia, Vancouver, BC V6T 2B5, Canada. ✉e-mail: akarsan@bcgsc.ca

hemogenic potential. Similarly, induction of *Gfi1* expression in HE cells promotes the emergence of IAHC from the vascular wall by repressing endothelial fate[11,13]. More recently, CD44 has also been identified as a marker of early hematopoietic fate[16]. Nevertheless, characterization of the processes occurring in the EC subpopulation that is primed to acquire a hemogenic fate remains challenging, and what drives the specification of EC towards an HE rather than a vascular fate is not well understood. Understanding the mechanisms that specify vascular EC towards a hemogenic fate is critical to improve bioengineering approaches to generate hematopoietic stem cells ex vivo.

While previous single-cell studies of EHT have relied on various markers to positively select very small and specific subpopulations[11,17,18], here we used CITE-seq to accurately label and comprehensively profile distinct subpopulations of cells in the AGM, therefore bypassing the need for isolation of rare cells by FACS. The combination of CITE-seq and bulk RNA-seq identified *Meis1* as a transcription factor required for hemogenic specification in the early embryo. *Meis1* is an important transcription factor for HSPC regulation in adult mouse bone marrow[19–23] and an essential developmental gene as evidenced by the embryonic lethality, between E11.5-E14.5, of mice with homozygous deletion of the locus[24,25]. *Meis1*-null fetal liver hematopoietic cells show impaired colony-forming activity and are unable to repopulate irradiated mice, suggesting that *Meis1*-null mice are unable to produce or maintain HSPC[25]. Although *Meis1* expression has been detected in emerging pre-HSPCs/HSPC during EHT[11,17,26,27], less is known about the function of *Meis1* at early stages of EHT. Here, we show that *Meis1* primes arterial EC towards the hemogenic fate, and that it is required in the mouse embryo for the formation of early pre-hemogenic cells prior to the emergence of functional HE and pre-HSPCs.

## Results

### CITE-seq provides a comprehensive representation of EHT
To define factors in the endothelium that prime cells towards a hemogenic phenotype, we performed CITE-seq on CD31⁺ FACS-isolated cells from the AGM of E10.5 wildtype (WT) embryos to isolate all EC and IAHC cells, rather than risk potential loss with additional pre-selected markers. Antibodies against CD44, CD117, CD41, CD43, CD27, and CD45 were used to identify populations of EC, HE, and pre-HSPC at various stages of the EHT process[1,3–6,16]. Unsupervised clustering of single-cell transcriptomes showed distinct populations of EC (expressing *Cdh5* (VE-Cadherin) and *Eng* (CD105, Endoglin)), cells undergoing EHT that expressed both endothelial and hematopoietic genes (ie: *Runx1*, *Gfi1*, *Spi1*), and mature hematopoietic cells (expressing *Ptprc* (CD45) and other lineage markers) (Fig. 1A, B). These mature hematopoietic cells, including erythro-myeloid progenitors (EMPs), are likely derived from early hematopoiesis in the yolk sac but may also represent a few mature cells generated at this stage from the EHT process. Multiple subpopulations of EC were present in the dataset and these were further classified based on expression of venous (vEC) and arterial (aEC) markers (Supplementary Fig. 1A, B).

In addition to EC and EHT clusters, the CITE-seq data revealed a small population of pre-hemogenic EC (pre-HE) similar to those recently described by Zhu et al.[18]. (Supplementary Fig. 1C, D). These cells followed a pseudotime trajectory towards EHT (Supplementary Fig. 1E), and had high expression of endothelial markers but additionally expressed *Cd44*, a marker recently associated with hemogenic potential[16], along with higher levels of *Procr* (CD201)[17] compared to other EC subsets (Fig. 1B and Supplementary Fig. 1F). Pre-HE cells lacked the prototypic HE markers *Runx1* and *Gfi1*, thus distinguishing these two populations (Fig. 1B and Supplementary Fig. 1G).

The additional immunophenotypic information provided by CITE-seq confirmed the expected subpopulations of EC, expressing negligible to no level of the selected surface markers, as well as cells undergoing EHT, which showed variable expression of CD117 and CD41 while upregulating CD43 and CD45 (Fig. 1C–E and

Supplementary Fig. 1H). Phenotypic HE cells expressing CD31⁺CD117⁻CD41⁻CD43⁻CD45⁻ and both *Runx1* and *Gfi1* were located at the extremity of the EHT cluster closest to pre-HE cells in UMAP space (Fig. 1F and Supplementary Fig. 1I). Of note, the presence of these HE cells represented a small fraction of the EHT cluster (13 out of 140 EHT cells), consistent with a potentially rapid transition to a pro-/pre-HSPC phenotype[18]. Most cells forming the EHT cluster were therefore IAHC, expressing both *Runx1* and *Gfi1* at the transcriptomic level and at least one phenotypic surface marker, CD117 or CD41 (Fig. 1E). In addition, we observed that the pre-HE cells of interest are indistinguishable from the remaining endothelium by surface protein expression of the typical phenotypic markers CD117 and CD41 (Fig. 1C, D). However, the addition of CD44 antibody showed selective enrichment for this small subset of pre-HE cells that we can now define as being CD31⁺CD44⁺CD117⁻CD41⁻CD43⁻CD45⁻ (Fig. 1C, D), and which represent an intermediate stage between the bulk population of aEC and cells undergoing EHT, as supported by in silico trajectory analysis (Supplementary Fig. 1E). Thus, CITE-seq identified distinct subsets of the endothelial and hematopoietic populations relevant to EHT together with single-cell gene expression data.

### Meis1 is upregulated in early pre-HE cells prior to EHT
Compared to aEC, 425 and 317 genes were upregulated and downregulated respectively in pre-HE (adjusted *p*-value < 0.1; Fig. 2A). The signature of upregulated genes was enriched for several pathways previously reported to be important for EHT such as TGF-β and Notch pathways, and inflammatory signaling (Supplementary Fig. 2A). In addition, gene ontology (GO) analysis identified the enrichment of genes related to hematopoiesis which further highlighted the hemogenic potential of these cells as compared to the aEC cluster (Supplementary Fig. 2A). Consistent with the pre-hemogenic nature of these cells, pre-HE maintained a high expression score of genes related to angiogenesis, compared to EHT cells and also showed a progressive upregulation of the definitive hemopoiesis gene signature (Supplementary Fig. 2B).

To look for drivers of cell fate, we focused on transcription factors (TFs). Of the 425 upregulated genes, 12 TFs were identified as being significantly enriched in pre-HE vs aEC and persisting through EHT (Fig. 2B). GO term analysis showed enrichment of these 12 TFs for biological processes associated with transcription and hematopoiesis (Fig. 2C). Therefore, we hypothesized that early expression of these TFs might govern the endothelial switch associated with hemogenic potency. Of the 12 TFs, *Meis1* ranked as being the most significantly enriched, with its binding partner *Pbx1* ranked fourth (Fig. 2B).

Separately, we performed bulk RNA-seq to identify upstream TFs present in the HE based on the rationale that these TFs might be important for driving pre-HE cells into the HE state. To this end, we used the *Runx1*+23GFP reporter mouse as a well-established tools to enrich for HE cells in the AGM[9]. A total of 228 transcription factors (TFs) were specifically enriched in CD31⁺CD117⁻CD45⁻*Runx1*+23GFP⁺ HE cells compared to vascular EC in the mouse AGM at E10.5. To find candidate upstream regulators that potentially drive the expression pattern observed in HE cells, Distant Regulatory Element (DiRE)[28] analysis was performed on the HE-enriched TFs, which resulted in a list of 9 enriched TFs that bind regulatory elements associated with co-expressed HE genes. These 9 TFs included *Meis1* and its binding partners *Pbx1* and *Hoxa9* (Supplementary Fig. 3). Therefore, two different approaches identified *Meis1* as enriched in the early stage of pre-HE/HE specification.

### Meis1 binds genes associated with hemogenic specification
Given the prediction that Meis1 can regulate the expression of other HE-enriched TFs in our dataset, we hypothesized that *Meis1* may be required to drive early hemogenic specification. Co-expression data from the EnrichR webserver[29] revealed that both *Meis1* and *Pbx1* were among the top TFs to co-occur with the 425 pre-HE-enriched genes (Fig. 2D), overlapping with 64 and 57 genes respectively.

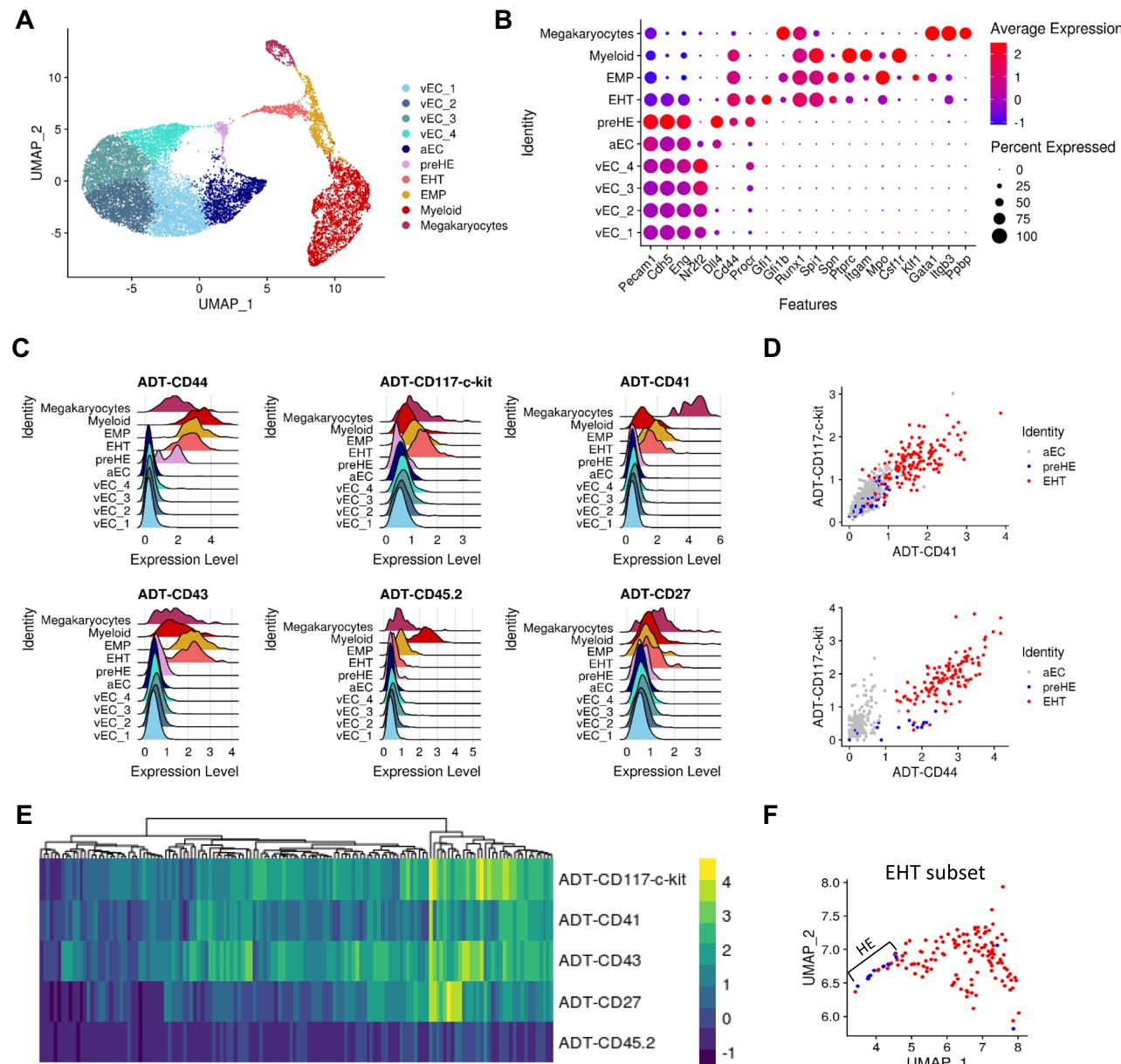

**Fig. 1 | CITE-seq characterization of CD31+ cells from E10.5 AGM. A** UMAP representation of single-cell unsupervised clustering based on transcriptome data. **B** Genes associated with each cluster in (**A**) that identify cell types. **C, D** Expression of antibody-derived tag (ADT) for (**C**) each cluster previously identified represented by ridge plots and **D** scatter plots of the relevant subpopulations. **E** Heatmap of surface expression, based on ADT, of the selected markers on cells forming the EHT cluster. **F** Identification of phenotypic HE within the "EHT" subcluster on a UMAP projection. Cells labeled in blue are negative for CD117, CD41, CD43, and CD45 based on ADT expression and red cells expressed at least one of the EHT markers.

We next performed ChIP-seq in primitive EPCR⁺CD150⁺CD48⁻lin⁻ adult hematopoietic cells transduced with a Meis1-YFP vector to identify direct Meis1 targets, resulting in binding peaks mapping to 16,198 nearby genes based on sequence proximity. We compared the list of genes associated with Meis1 binding sites to two publicly available ChIP-seq datasets based on endogenous Meis1 expression: one performed in HPC7 mouse hematopoietic progenitor cells[30] and the other in E11.5 mouse embryos[31]. Genes were considered bound by Meis1 if they were found in common between at least two of the three datasets. Over 59% (252/425) of the genes upregulated in pre-HE cells compared to aEC were associated with Meis1 binding sites (Fig. 2E). These genes included 11 of the 12 hemogenic TFs that we previously identified in pre-HE compared to aEC, placing *Meis1* as a major candidate upstream regulator. Promoters of 52 of the 64 genes frequently

co-expressed with *Meis1* based on EnrichR analysis were bound by Meis1 based on ChIP-seq, supporting a role for Meis1 in driving aEC towards a hemogenic fate.

Compared to aEC, the pre-HE gene signature was enriched for GO terms related to hematopoiesis (Supplementary Fig. 2A). Interestingly, GO term analysis showed an association of biological processes related to the development of the hematopoietic system only in the fraction of pre-HE genes that are predicted to be bound by Meis1 (Fig. 2F). On the other hand, genes related to angiogenesis and developmental processes were more broadly distributed between the Meis1-bound and unbound fractions (Fig. 2F). Thus, genes related to the development of the hematopoietic system that are expressed in pre-HE cells are also direct Meis1 targets, suggesting that Meis1 primes aEC for hemogenic commitment by acting as an upstream regulator of hematopoietic

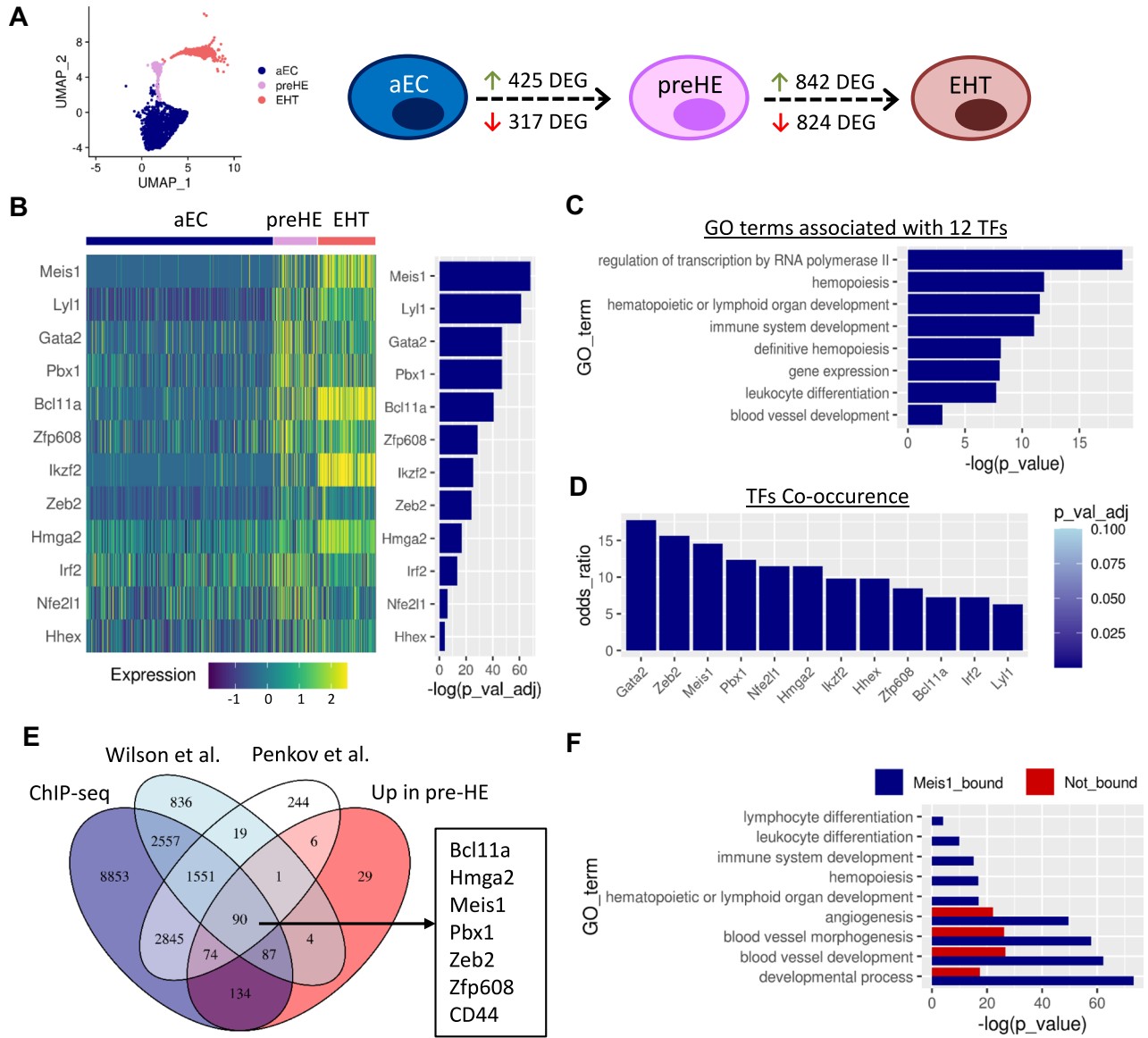

**Fig. 2 | Identification of Meis1 as candidate regulator of early hemogenic commitment. A** UMAP representation of the aEC, preHE, and EHT clusters only and schematic showing the number of DEG between each cell population along the transition. **B** Heatmap showing expression of the 12 TFs enriched in pre-HE cells compared to aEC, ranked by significance (Wilcoxon rank sum test (two-sided) with Bonferroni correction for adjusted *p*-values). **C** Selected GO terms associated with the 12 TFs identified in **B** (hypergeometric p-value with g:SCS correction using the default settings in gprofiler2). **D** Co-occurrence of the 12 TFs in **B** with the 425 pre-HE enriched genes, based on EnrichR analysis (*p*-values based one-sided Fisher's exact test with Benjamini-Hochberg correction). **E** Overlapping genes between our Meis1 ChIP-seq dataset, published ChIP-seq datasets, and genes enriched in pre-HE vs aEC in the CITE-seq data. **F** Selected GO terms associated with pre-HE enriched genes categorized based on the presence of Meis1 binding sites in ChIP-seq data (hypergeometric p-value with g:SCS correction using the default settings in gprofiler2).

gene expression. Several genes downregulated during the transition to pre-HE are also found in proximity to Meis1 binding sites (Supplementary Fig. 4). While these genes enriched for more generalized biological GO terms, the list included angiogenic and Wnt-related genes that need to be downregulated for EHT to proceed (Supplementary Fig. 4). Meis1 may also have a potential repressive role on endothelial fate through a few key genes.

#### *Meis1* is highly expressed in IAHC and the surrounding endothelium in vivo

To validate the role of *Meis1* in hemogenic EC specification, we used a *Meis1* GFP-reporter mouse model (Fig. 3A) where a GFP-P2A-HA tag is inserted upstream of the *Meis1* transcriptional start site[32]. GFP

faithfully recapitulates Meis1 expression without altering its function due to a 2A-self-cleavage reaction during translation (Supplementary Fig. 5A, ref. 32). In the E10.5 AGM, GFP was highly expressed in all IAHC and in a subset of EC of the dorsal aorta, as well as in the subaortic mesenchyme (Fig. 3B). Consistent with a role in IAHC emergence, GFP⁺ EC were enriched in sections of the dorsal aorta underlying IAHC as opposed to GFP⁻ EC that could be seen on long sections devoid of IAHC (Fig. 3B). Similarly, analysis by flow cytometry confirmed ubiquitous high expression of *Meis1* in phenotypic pre-HSPC (CD31⁺CD41^low^CD45^−/+^) (Fig. 3C, D), as observed in the CITE-seq data (Fig. 3E). Consistent with the immunostaining, most EC (CD31⁺CD41⁻CD45⁻) were GFP⁻ by flow cytometry with a distinct smaller subset of GFP⁺ EC (Fig. 3C). Within the bulk EC compartment,

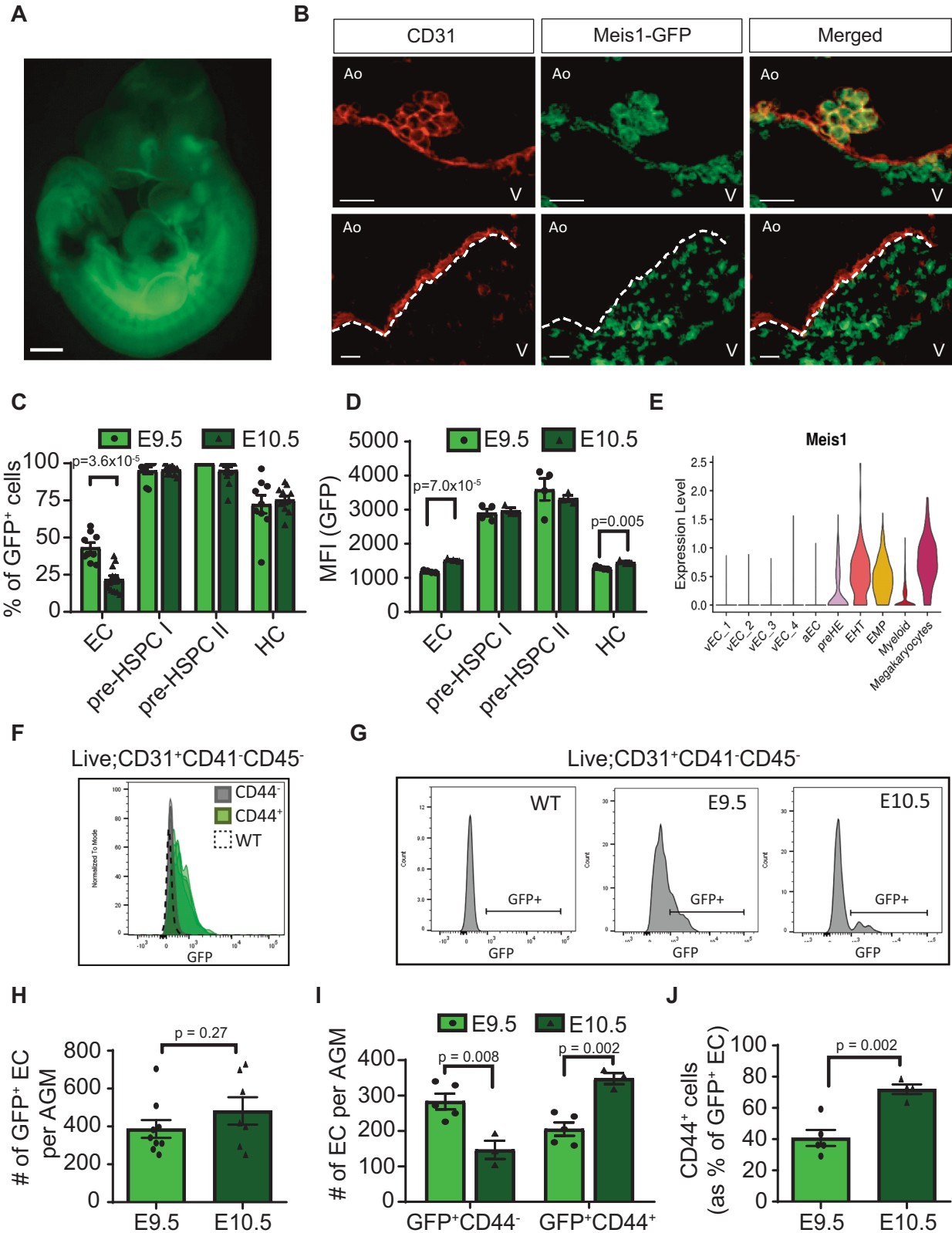

GFP was almost exclusively expressed in CD44$^+$ EC confirming that *Meis1* labeled the CD44$^+$ pre-HE fraction of the endothelium at E10.5 (Fig. 3F), thus validating at the protein level in vivo the *Meis1* expression pattern observed at the transcript level in the CITE-seq data at E10.5 (Fig. 3E).

While IAHC are prominent in the E10.5 dorsal aorta, specification of the HE precedes cluster formation. Tracking Meis1 expression at an earlier time point showed a relatively higher proportion of GFP$^+$ cells at E9.5 compared to E10.5 specifically within the EC compartment (Fig. 3C, G). Although, no difference was observed in absolute GFP$^+$ EC number between the two time points (Fig. 3H), E10.5 GFP$^+$ EC showed increased Meis1 expression with the initiation of EHT (Fig. 3D, G). CD44 expression temporally followed Meis1 expression in GFP$^+$ EC as suggested by the increased numbers of CD44$^+$ cells in the GFP$^+$ EC from

**Fig. 3 | Meis1 expression in the AGM during EHT. A** Representative image of more than 50 *Meis1*-GFP embryos dissected at E10.5 (scale bar = 500 μm). **B** Immunostaining of cross-sections of the dorsal aorta at E10.5 showing the expression of CD31 (red) and Meis1-GFP (green). The dotted line in the lower panels demarcates the endothelial layer from the subaortic mesenchyme. (Ao: Aorta, V: Ventral, scale bar = 20 μm). N = 7 embryos over 4 independent experiments. **C** Percentage of cells expressing Meis1 in defined cell populations based on GFP fluorescence by flow cytometry (n = 9 at E9.5 and n = 12 at E10.5) and (**D**) associated GFP MFI values (n = 4 at E9.5 and n = 3 at E10.5 from the same experiment). Viable cells were gated as follows: EC: CD31$^+$CD41$^-$CD45$^-$, pre-HSPC I: CD31$^+$CD41$^{low}$CD45$^-$, pre-HSPC II: CD31$^+$CD41$^{low}$CD45$^+$, hematopoietic cells (HC): CD31$^-$CD45$^+$. Data are presented as mean ± SEM and each data point represents a different embryo. **E** Violin plot depicting *Meis1* mRNA expression in various CITE-seq subpopulations.

*Meis1* expression was detected at low level in 9.9% of all EC. **F** Histograms showing *Meis1*-GFP expression by flow cytometry in CD44$^-$ and CD44$^+$ EC (n = 4 for each group). WT represents a wildtype (GFP$^-$) embryo. **G** Representative histograms of GFP fluorescence by flow cytometry in AGM EC at various time points. **H** Total number of GFP$^+$ EC per AGM at E9.5 (n = 9) and E10.5 (n = 7) and (**I**) EC numbers subdivided according to CD44 co-expression (n = 5 at E9.5, n = 3 at E10.5) based on counting beads by flow cytometry. For the absolute cell number, one E10.5 data point was removed based on the Grubb test to identify outliers. **J** Proportion of GFP$^+$ EC co-expressing CD44 at E9.5 (n = 5) and E10.5 (n = 4). Data are presented as mean ± SEM and each data point represents a different embryo. All indicated p-values were calculated using a two-sided t-test comparing two groups. Source data are provided as a Source Data file.

E9.5 to E10.5 (40% vs 72%, Fig. 3I, J). This temporal increase in frequency of CD44$^+$ cells implicates a progression from an early GFP$^+$CD44$^-$ precursor at E9.5 to GFP$^+$CD44$^+$ pre-HE cells at E10.5, given that the absolute number of GFP$^+$ EC between the two time points is similar. Supporting this point, E9.5 GFP$^+$CD44$^-$ cells have already upregulated pre-HE genes, that are also Meis1 targets, as compared to GFP$^-$CD44$^-$ EC but do not yet express HE genes (Supplementary Fig. 5B, C). These findings suggest that *Meis1* is an early driver of hemogenic specification.

### Runx1$^+$ HE cells emerge from Meis1$^+$ endothelium in vivo

The observed expression pattern argues that only the *Meis1*-GFP$^+$ EC have hemogenic potential and that *Meis1* might be important in the early specification of CD44$^+$ pre-HE cells. To investigate this, GFP$^-$ EC and GFP$^+$ EC were FACS-isolated from dissociated E10.5 AGM to further characterize these two populations (Fig. 4A and Supplementary Fig. 8A). Consistent with our hypothesis, expression of the EHT-related genes *Runx1*, *Gfi1*, and *Spi1* was exclusively detected in GFP$^+$ EC (Fig. 4B), indicative of the presence of HE cells only within the *Meis1*-expressing population. While *Runx1* expression was entirely contained within *Meis1*-GFP$^+$ EC (Fig. 4B), *Meis1* was more broadly expressed and detected at lower levels in *Runx1*+23GFP$^-$ EC from the AGM at E10.5 (Fig. 4C). This observation was consistent with the CITE-seq data (Fig. 4D) and reinforces the fact that *Meis1*-expressing EC include both pre-HE and HE cells, and precede *Runx1* expression (Fig. 4D). Fluorescent imaging confirmed that Meis1 expression was detected in all emerging Runx1$^+$ HE in vivo, but also in Runx1$^-$ EC in an extended region flanking Runx1$^+$ HE (Fig. 4E and Supplementary Fig. 6), consistent with the idea that HE cells arise from a *Meis1*-expressing EC precursor in vivo.

Neither *Meis1*-GFP nor CD44 individually are able to separate pre-HE cells from HE cells. Ideally, a combination of surface markers would distinguish between and facilitate the isolation of pre-HE from HE. Based on our CITE-seq data and a published dataset[18], we identified two potential candidate surface molecules, CD200 and Notch4, that are expressed in EC and pre-HE but not in HE and EHT cells (Supplementary Fig. 7A). Compared to E10.5 CD44$^-$ EC, cells co-expressing CD44 and either CD200 or Notch4 showed higher levels of genes upregulated in the CITE-seq pre-HE population (Supplementary Fig. 7B–E and Supplementary Fig 5C). On the other hand, HE genes were upregulated in CD44$^+$CD200$^-$ and CD44$^+$Notch4$^-$ cells and more strongly expressed in pre-HSPC (Supplementary Fig. 7F, G). Unlike *Runx1*, *Meis1* is strongly expressed in both CD44$^+$CD200$^-$ or CD44$^+$Notch4$^-$ and CD44$^+$CD200$^+$ or CD44$^+$Notch4$^+$ cell fractions, consistent with a requirement for Meis1 in pre-HE prior to *Runx1* upregulation during early EHT.

To demonstrate functionally that only *Meis1*-GFP$^+$ cells have the potential to undergo EHT, E10.5 GFP$^+$ and GFP$^-$ EC were co-cultured ex vivo on OP9 feeder cells in media with cytokines supportive of hematopoiesis (Fig. 4F and Supplementary Fig. 8B). CD41$^+$ pre-HSPC and CD45$^+$ hematopoietic cells were included as controls (Supplementary Fig. 8A, B) to confirm that this culture system can promote

differentiation of CD31$^+$CD41$^{low}$CD45$^-$ cells into CD45$^+$ cells and maintain them in culture. Colonies of round CD45$^+$ hematopoietic cells emerged from CD31$^+$CD41$^-$CD45$^-$GFP$^+$ EC at 5 days, but were totally absent in wells containing GFP$^-$ EC (Fig. 4F) regardless of the initial number of cells plated, implying that expression of *Meis1* is required to generate functional HE cells. Importantly, the *Meis1* gene locus is genetically unaltered in this model, therefore GFP$^-$ cells still have the potential to spontaneously upregulate *Meis1* expression if required at a later stage during the culture. The fact that none of the GFP$^-$ EC subsequently formed hematopoietic colonies ex vivo in differentiation-permissive medium supports the contention that *Meis1* activation is required at early stages of EHT, prior to the formation of pre-HSPCs.

### Meis1 expression is required for pre-HE/HE formation

To investigate a functional requirement for *Meis1* in EC, we conditionally deleted *Meis1* using a Cre recombinase driven by the VE-Cadherin (VEC) promoter[33] to generate *Meis1*$^{flox/flox}$;Cre (M$^{f/f}$;Cre) embryos (Supplementary Fig. 9) and littermates lacking Cre expression hereafter referred to as "WT". To quantify the frequency of HE cells, we co-cultured with OP9 FACS-isolated EC from single embryos into multiple wells of a 96-well plate, at a limiting dilution of 100 cells per well (Fig. 5A). Significantly fewer cells from E9.5 M$^{f/f}$;Cre embryos produced hematopoietic colonies compared to WT littermates (Fig. 5B), supporting a requirement for *Meis1* in emergence of functional HE. Confirming that the defect in *Meis1*-null EC was present at the HE level, gene expression analysis performed on FACS-isolated EC showed a significant reduction in key markers of HE cells (*Runx1*, *Gfi1*, and *Spi1*) (Fig. 5C), expressed exclusively in *Meis1*-GFP$^+$ EC (Fig. 4B).

To better characterize the origin of the defect at the single cell level, CITE-seq was performed on E10.5 M$^{f/f}$;Cre embryos to compare the different subpopulations to their WT counterpart. A total of 6745 M$^{f/f}$;Cre cells and 8423 WT cells (Supplementary Table 2) were integrated and visualized by dimensional reduction of transcriptomes on UMAP plots (Fig. 5D). Significant differences in sample distribution were observed with M$^{f/f}$;Cre cells being significantly under-represented in both the pre-HE and EHT subsets (Fig. 5E and Supplementary Table 2). Focusing on the subpopulations of interest confirmed that the proportion of pre-HE and EHT cells were significantly reduced compared to aEC in M$^{f/f}$;Cre samples (Fig. 5F). Interestingly, the relative proportion of pre-HE progressing to EHT cells was not significantly different from what was observed in WT samples (Fig. 5F). Therefore, the reduction observed in the number of cells undergoing EHT resulted from the initial early decrease in the pre-HE population, with a concomitant increase in aEC, and not a blockade in the pre-HE to EHT transition per se. Consistent with the observation of a defect in the generation of pre-HE, M$^{f/f}$;Cre embryos showed a reduction in the proportion of EC that expressed CD44 at E9.5 (Fig. 5G), corroborating the findings that *Meis1* contributes to the specification of early pre-HE cells.

Interestingly, while some pre-HE cells are formed in the absence of *Meis1*, they appeared to be less transdifferentiated. We used the gene

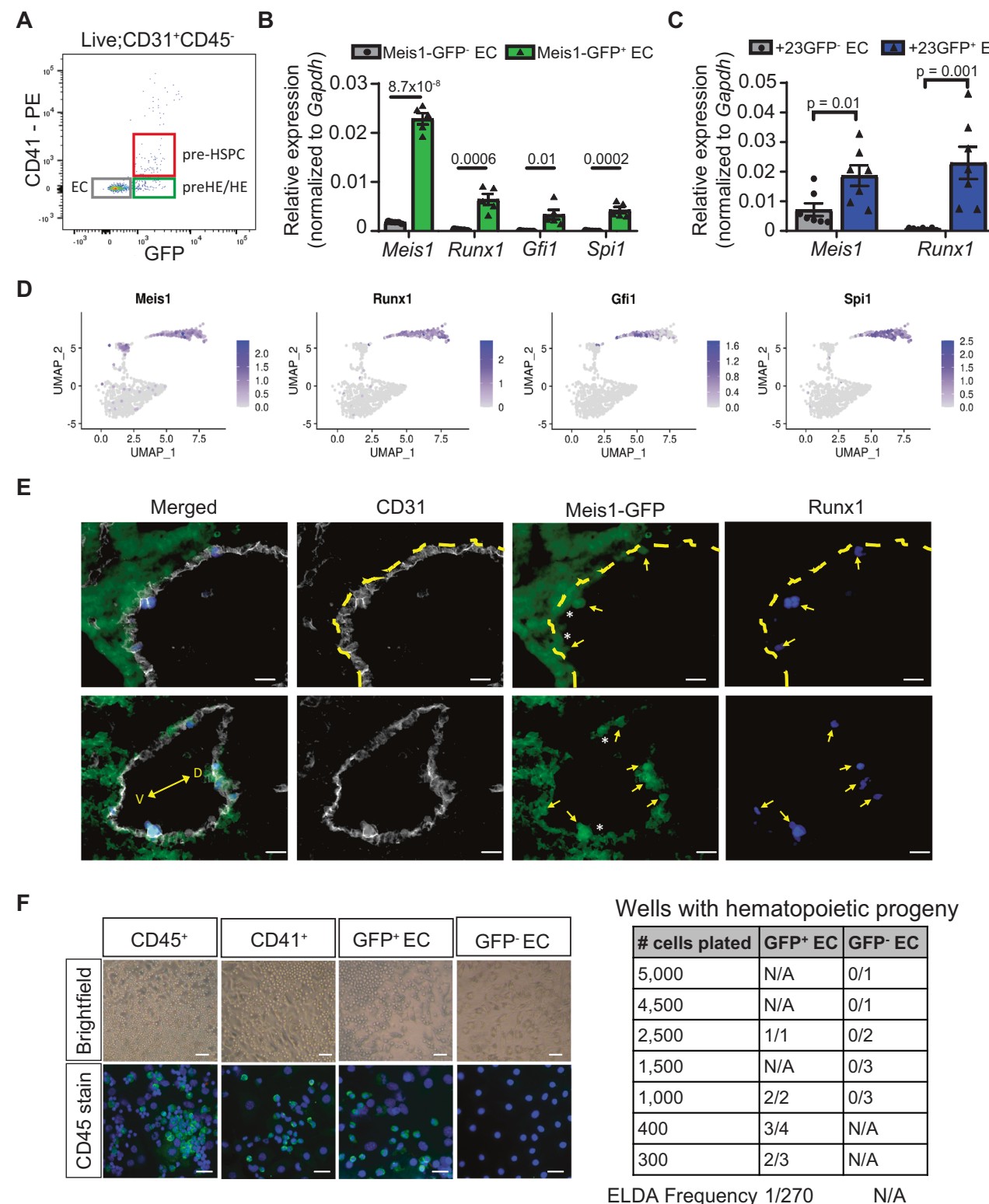

signatures of aEC and pre-HE populations, based on differential analysis between these two clusters in WT cells (Fig. 2A), to compute an expression score for each WT and M^f/f;Cre cell contained in the pre-HE cluster. Overall, WT pre-HE cells had higher expression of genes upregulated in pre-HE vs aEC whereas M^f/f;Cre cells scored higher for genes that would be expected to be downregulated in that cluster (Fig. 5H). Therefore, *Meis1* plays a critical role in the acquisition of a pre-HE transcriptional program in cells that would otherwise become aEC.

**Aberrant pre-HE formation impairs the emergence of pre-HSPCs**

Consequent to the loss of *Meis1* expression, the reduced hemogenic potential of M^f/f;Cre EC from the dorsal aorta impaired the emergence of pre-HSPC. Flow cytometry showed a lower number of immunophenotypic CD31⁺CD41^low CD45⁻ pre-HSPC in the AGM of M^f/f;Cre embryos both as a proportion of EC and by absolute cell counts at E10.5 (Fig. 6A, B). Similarly, fewer immunophenotypic pre-HSPC were detected in E9.5 M^f/f;Cre AGM compared to WT littermates (Fig. 6C). While the number of WT pre-HSPC strongly increased between E9.5

**Fig. 4 | Runx1+ HE emerge from Meis1-expressing pre-HE in vivo.**
**A** Representative flow cytometry scatterplot gated for relevant subsets within the viable CD31⁺CD45⁻ population. **B, C** mRNA expression of selected EHT-related genes in FACS-isolated GFP⁻ and GFP⁺ EC from E10.5 AGM of (**B**) *Meis1*-GFP embryos ($n = 5$ pools of 4 embryos each from the same litter) and (**C**) *Runx1*+23GFP embryos ($n = 7$ pools of 3 embryos each). Quantification was performed by ddPCR and normalized to Gapdh. Bar-plots are showing the mean ± SEM. Indicated *p*-values were calculated using a two-sided t-test comparing two groups. **D** UMAP projections of the aEC, pre-HE, and EHT clusters showing the expression of selected genes in the CITE-seq data. **E** Immunostaining of the dorsal aorta from E10.5 *Meis1*-GFP embryos (gray: CD31, green: GFP, blue: Runx1, V: ventral, D: dorsal). Arrows point to

Runx1+ cells and asterix (*) denotes adjacent Meis1-GFP⁺Runx1⁻ EC. The dotted line delineates the endothelial layer from the subaortic mesenchyme in upper panels. Scale bar = 20 µm. Staining was reproduced in 2 independent experiments. **F** Ex vivo differentiation assay on an OP9 stromal layer to assess hematopoietic colony formation from FACS-isolated GFP⁺ and GFP⁻ EC. Representative brightfield images (above, scale bar = 50 µm) of hematopoietic colonies formed and confirmation of CD45 expression by immunofluorescence staining (below, scale bar = 25 µm). The table on the right shows the number of colony-positive replicates at the endpoint for the various initial cell inputs tested. The frequency of HE cells was assessed using extreme limiting dilution assay (ELDA) calculation. Source data are provided as a Source Data file.

and E10.5, the fold reduction observed in M^f/f;Cre embryos compared to WT was maintained over the two time points (2.23-fold vs 2.37-fold reduction at E9.5 and E10.5, respectively). The EHT defect thus originates early in development and is maintained and propagated over the developmental process. As the defect is not amplified over time, this suggests that Meis1 is essential for early pre-HE specification, but not for promoting hematopoietic transdifferentiation to early pre-HSPC.

To further confirm that the lack of pre-HE cells in the absence of *Meis1* expression reduces EHT in vivo, we backcrossed the *Meis1* conditional-knockout mice with the *Runx1*+23GFP reporter mouse to generate transgenic embryos that expressed GFP under the spatiotemporal control of the *Runx1*+23-enhancer, thus marking all emerging functional pre-HSPC[9], and lacking EC expression of *Meis1* (Meis1^flox/flox;VEC-cre;+23GFP, abbreviated M^f/f;Cre;+23GFP). Not every litter produced these triple-transgenic embryos therefore, embryos were collected to examine IAHC by immunostaining cross-sections of the dorsal aorta using the +23GFP as a marker. Some +23GFP cells were detected in both WT (M^f/f;+23GFP or M^f/+;+23GFP) and M^f/f;Cre;+23GFP dorsal aortas. However, the number of large IAHC ( > 4 cells per cluster) was significantly reduced in M^f/f;Cre;+23GFP dorsal aortas (Fig. 6D, E). Most +23GFP cells remained flat and embedded in the vascular wall of the dorsal aorta as opposed to bulging into the vascular lumen as observed in WT counterparts (Fig. 6D). Lack of *Meis1* prevented pre-HE specification and *Meis1*-null cells appeared unable to undergo EHT despite expression of the *Runx1*+23GFP transgene. As with pre-HSPC numbers quantified by flow cytometry, IAHC formation in *Meis1*-deficient embryos was reduced by 2.38-fold secondary to the defect in specification of vascular EC to pre-HE. Even though some IAHC formed, these M^f/f;Cre EHT cells were more immature than their WT counterpart based on single-cell gene expression signatures (Fig. 6F).

### *Meis1* induces a pre-HE transcriptional state in adult EC
The ability of *Meis1* to prime EC towards a hemogenic transcriptional program was assessed by introducing *Meis1* via lentiviral transduction into aortic EC from adult C57BL/6 mice, followed by isolation of transduced cells that co-expressed CD31 and VEC to ensure that the cells maintained an EC phenotype 72 h post-transduction (Fig. 7A). An increase in surface expression of CD44 was detectable in *Meis1*-over-expressing cells (Meis1-OE) (Fig. 7B), compatible with the temporal upregulation of CD44 observed between E9.5 and E10.5 GFP⁺ EC in vivo (Fig. 3I, J), and places *Meis1* as a regulator upstream of CD44.

RNA-seq of Meis1-OE cells showed 1205 upregulated and 1300 downregulated genes (Fig. 7C). As predicted, cells that overexpressed *Meis1* became more similar to pre-HE. Significant overlap was observed between genes upregulated following Meis1-overexpression and genes enriched in pre-HE cells compared to aEC (Fig. 7D). Specifically, 56 of the 425 pre-HE-enriched genes were found upregulated in Meis1-OE EC, which was significantly more than expected by random sampling ($p = 5.01e-05$, Fig. 7D). Gene set enrichment analysis performed using custom genesets that included all the DEGs between aEC and pre-HE confirmed that Meis1-OE EC were significantly enriched in genes upregulated in pre-HE vs aEC (Fig. 7E). In addition, genes down-regulated upon Meis1-OE were enriched in genes downregulated in

pre-HE vs aEC (Fig. 7E). Therefore, expression of Meis1 alone in non-hematopoietic adult EC was able to trigger transcriptional changes reminiscent of the expression pattern that we observed in embryonic pre-HE, demonstrating a clear role for *Meis1* in inducing a cellular program that is a prerequisite for reaching hemogenic competency.

While a clear pre-HE transcriptional program was detected in *Meis1*-expressing EC, adult EC that overexpressed *Meis1* failed to form hematopoietic cells in vitro under the permissive OP9 co-culture conditions. After 6 days of co-culture, most cells retained VEC expression and no change in relative expression of the surface markers CD117, CD43, or CD45 could be detected when compared to YFP-CTRL cells (Fig. 7F, G and Supplementary Fig. 10). Nonetheless, the upregulation of CD44 previously observed after 72 h (Fig. 7B) was even more pronounced after an additional 6 days of co-culture in the presence of cytokines (Fig. 7F–H and Supplementary Fig. 10), further validating the positive regulation of CD44 by Meis1. While Meis1 pushes EC towards the pre-HE state, it appears to be insufficient to drive cells all the way through EHT, at least in the culture conditions used in this system. Meis1 induces a pre-HE-like program in EC, but it likely requires additional transcriptional support to facilitate EHT. Indeed, this finding provides a corollary to the results in Fig. 5, which suggest that Meis1 is not required for the initiation of EHT once a pre-HE state is achieved. Taken together, our findings imply that the main function of *Meis1* in early blood development appears to coax aEC into the pre-HE state, and that it is neither necessary nor sufficient to further promote EHT in committed HE cells.

## Discussion
The formation of HSPC requires a specialized endothelium with the potential to undergo EHT. Defining the characteristics of these EC capable of progressing to an HE fate is required to improve the in vitro generation of blood cells from alternative sources such as iPSC or even adult endothelium[34,35]. As a precursor for the generation of HE and EHT, the recently identified pre-HE population[18] requires better characterization. Using a single-cell sequencing approach to dissect the heterogeneity within the endothelium, our data corroborates the existence of a pre-HE cell population that is distinct from both aEC and traditionally defined HE cells. As such, the traditional reporters used to enrich for HE cells[9,11,36] are activated after this pre-HE subset has differentiated, thus making these cells largely uncharacterized. The use of oligo-conjugated antibodies in our CITE-seq experiment defined these pre-HE cells as CD31⁺CD44⁺CD117⁻CD41⁻CD45⁻, consistent with their endothelial phenotype and in agreement with the recent proposition that CD44^low CD117⁻ mark VEC⁺ EC with EHT potential[16]. Here, we identify *Meis1* as a critical regulator of pre-HE cell specification, acting upstream of key hemogenic-associated genes including *Runx1* and the surface marker CD44.

Importantly, GFP-positive cells in a *Meis1*-GFP reporter mouse[32] label and isolate the EC population enriched for pre-HE and HE cells. Using the *Meis1*-GFP reporter mouse and *Runx1* expression, we show that Meis1⁺ pre-HE cells reside adjacent to emerging Runx1⁺ cells and IAHC (Fig. 4E). Consistent with the notion that polyclonal IAHC are formed via recruitment of adjacent cells[37], the localization of Meis1⁺

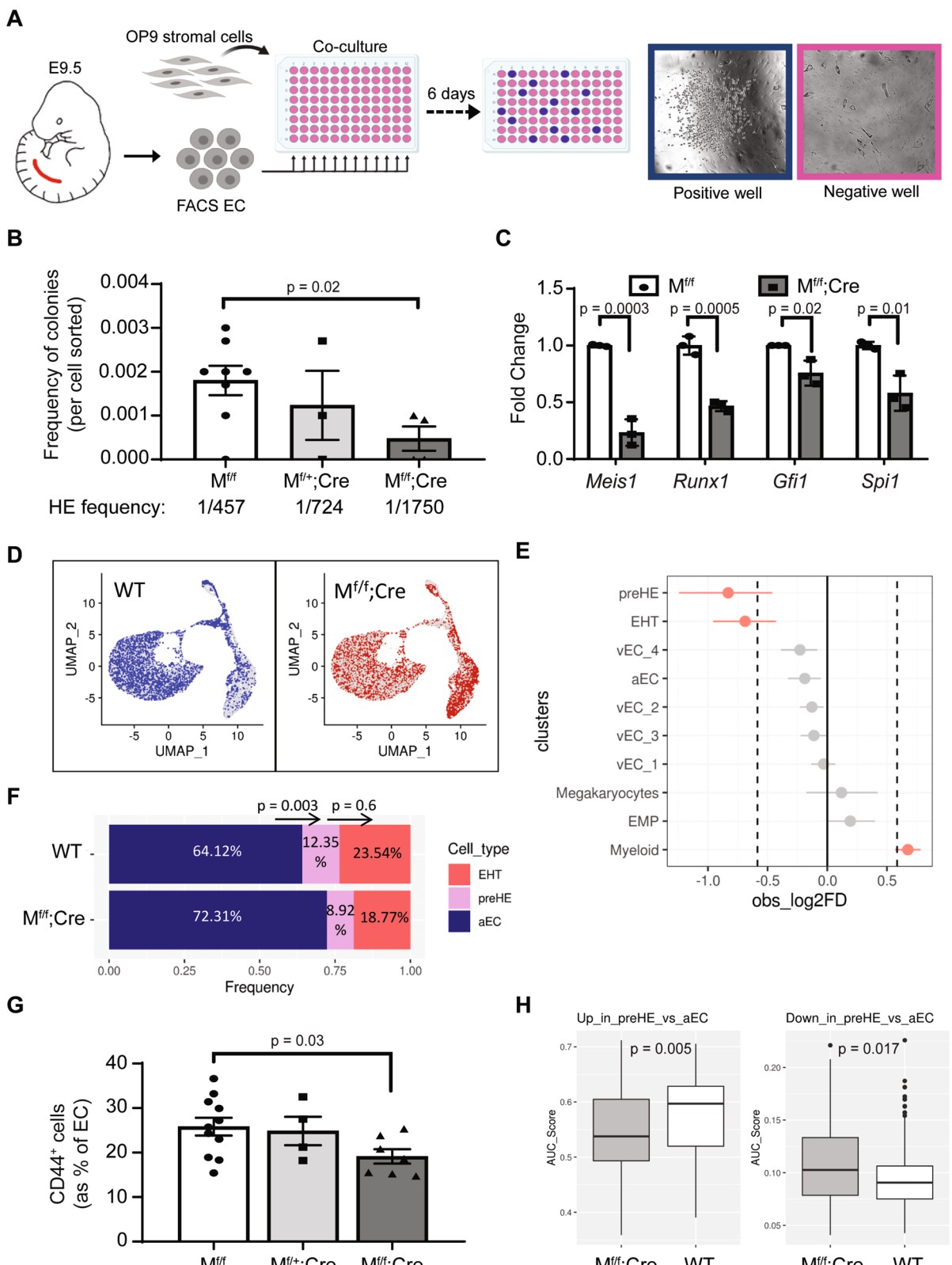

pre-HE further supports their contribution to IAHC emergence. Combined with a *Runx1* reporter or the published *Gfi1*[+/tomato] mouse[13], this *Meis1*-GFP mouse could provide a much needed genetic tool to distinguish or track the populations of aEC, pre-HE, and HE cells in future studies. Alternatively, we also identified two surface markers, CD200 and Notch4, that could be used in combination with current markers such as CD31, CD44, and CD41 to potentially isolate pre-HE

from HE cells. However, further functional validation will need to be performed on these subpopulations.

Conditional *Meis1*-knockout mice reveal that *Meis1* expression in VEC-expressing EC is required to directly govern the fate of these EC in vivo in the mouse embryo. Indeed, the population of pre-HE is significantly reduced in the absence of *Meis1* in VEC[+] cells. On the other hand, enforced expression of *Meis1* alone in *Meis1*-negative adult

**Fig. 5 | Deletion of endothelial *Meis1* affects pre-HE formation and decreases hemogenic potential of the dorsal aorta. A** Schematic representation of the ex vivo co-culture assay used to quantify the frequency of HE cells (created with BioRender.com) and **B** frequency of functional HE cells in E9.5 AGM ($n = 8$ M$^{f/f}$, 3 M$^{f/+}$;Cre, 4 M$^{f/f}$;Cre). Each dot in **B** represents the outcome of 1 embryo and the average frequency of HE cells was approximated using the Extreme Limiting Dilution Assay (ELDA) software[48]. **C** mRNA expression of selected EHT-related genes in FACS-isolated viable EC (CD31$^+$CD41$^-$CD45$^-$) from E10.5 AGM. Quantification was performed by ddPCR and normalized to Gapdh ($n = 3$ independent replicates per genotype). In **B** and **C**, data are presented as mean ± SEM and p-values were calculated using a two-sided t-test comparing two groups. **D** Representation of WT (blue) and M$^{f/f}$;Cre (red) cells in UMAP space based on unsupervised clustering. Cells from the opposite genotype are represented in gray. **E** Analysis of the difference between the proportion of WT and M$^{f/f}$;Cre cells found in each cluster of the CITE-seq data, using scProportionTest package (permutation test; $n = 1000$). The populations were considered over-/under-represented at a cutoff fold-difference > |

1.5| (log2FD > |0.58|; dotted line) and significant based on FDR < 0.05 (red dots). Data are presented as mean ±95% confidence interval ($n = 8423$ WT cells from five independent samples and $n = 6745$ M$^{f/f}$;Cre cells from three independent samples). **F** Relative proportions of aEC, pre-HE, and EHT cells in WT and M$^{f/f}$;Cre samples. A two-proportion Z-test (two-sided) was performed comparing the cell ratio in adjacent clusters to assess the difference between genotypes in cells transitioning from aEC to pre-HE and pre-HE to EHT (p-values are displayed above the bars). **G** Quantification of CD44$^+$ pre-HE cells in E9.5 AGM by flow cytometry ($n = 11$ WT, 4 M$^{f/+}$;Cre, and 7 M$^{f/f}$;Cre). Data are presented as mean ± SEM and each data point represents a different embryo (p-value based on two-sided t-test). **H** Gene expression scores for pre-HE gene signatures (up (left panel) or down (right panel)) in individual pre-HE cells from WT and M$^{f/f}$;Cre samples, calculated using AUCell. Box-plots display the median (center line), the 25th and 75th percentiles (box limits), and ±1.5*interquartile range (whiskers). $n = 171$ WT cells from five independent samples and $n = 77$ M$^{f/f}$;Cre cells from three independent samples. P-values were calculated using two-sided t-test. Source data are provided as a Source Data file.

arterial EC induced many transcriptional changes that also supports a critical requirement for *Meis1* in the transition from aEC into pre-HE. Together, our findings reveal a critical function for *Meis1* at the onset of EHT, in addition to its known roles in HSPC maintenance and hematopoiesis in general.

## Methods

All mouse experiments were conducted following review and approval by the UBC Animal Care Committee.

### Mice

All breeding and experiments were conducted in British Columbia Cancer Research Animal Resource Centre barrier facility. Mice were housed with a 14/10-hour light/dark cycle, with free access to food and water. Breeding and experimental rooms were maintained at an ambient temperature of 20-26 °C, and a humidity of 40–60%. *Meis1*-GFP mice (C57BL/6 background)[32] were maintained as heterozygous and mated with WT C57BL/6. *Meis1*$^{fl/fl}$ mice with loxP sites flanking the exon 8 were originally generated by Drs Nancy Jenkins and Neal Copeland[19,22,23]. VEC-Cre (B6;129-Tg(Cdh5-cre)1Spe/J) males were purchased from The Jackson Laboratory[33] and crossed with *Meis1*-floxed animals twice to generate embryos with homozygous excision for experiments. The Cre allele was carried by heterozygous males to allow for consistent tracking of Cre efficiency by repeated mating of each stud. *Meis1*$^{fl/fl}$;*Runx1*+23GFP animals were generated in-house by crossing our *Meis1*-floxed mice with *Runx1*+23mCNE-GFP mice generously provided by Dr. Motomi Osato[15]. These females were then used in timed matings with *Meis1*-VEC-Cre males. Embryos were collected at specific time points following detection of a vaginal plug (E0.5) and genotyped using genomic DNA extracted from the yolk sac by digestion with proteinase K, followed by PCR using specific primers (Supplementary Table 3). All animal protocols were approved by the Animal Care Committee of the University of British Columbia (Vancouver, Canada).

### Cell lines

OP9 cells were purchased from ATCC (cat# CRL-2749) and cultured in alpha Modified Eagle Medium (α-MEM), supplemented with 10% heat-inactivated fetal bovine serum (FBS), 2 mM glutamine, and 100 U each of penicillin and streptomycin. C57BL/6 mouse primary aortic EC were purchased from Cell Biologics (cat# C57-6052) and cultured in MCDB 131 medium supplemented with 20% FBS, 2 mM glutamine, 100 U each of penicillin and streptomycin, 100 μg/ml heparin, and 50 μg/ml ECGS.

### Flow cytometry

Dissected AGM were dissociated in PBS with 2% fetal bovine serum, 1 mg/ml collagenase II, and 1 mg/ml DNaseI. Cell were resuspended in

PBS-2%FBS with DNaseI, blocked 5 min with anti-mouse CD16/32 (BD Biosciences #553142; 7 μl/ml) and stained with the following antibodies for 30 min, protected from light: Anti-Mouse CD45.2 [104] APC-eFluor® 780 (ThermoFisher #47-0454-82; 1:400 dilution) or Anti-mouse CD45 [30-F11] PE/Cy7 (BioLegend #103113; 1:400), Anti-Mouse CD31 (Pecam1) [390] BV605 (BioLegend #102427; 1:400), Anti-Mouse CD41 [MWReg30] PE (ThermoFisher #12-0411-82; 1:400), Anti-Mouse CD41 [MWReg30] FITC (BioLegend #133904; 1:400), or Anti-Mouse CD41 [MWReg30] APC/Cy7 (BioLegend #133928; 1:400), Anti-Mouse/Human CD44 APC (BioLegend, cat# 103012; 1:400), Anti-mouse Notch4 [HMN4-14] PE (ThermoFisher #12-5764-80; 1:400), Anti-mouse CD200 [OX90]PerCPe710 (ThermoFisher #46-5200-80; 1:400), Anti-mouse CD144 [VECD1] PE (BioLegend #138105; 1:400), Anti-mouse CD43 [S11] PE/cy7 (BioLegend #143209; 1:400), and Anti-mouse CD117 [2B8] BV650 (BD Biosciences #563399; 1:400). Cells were sorted on a BD FACSAria III or BD FACSAria Fusion instrument and data was analyzed using FlowJo software.

### Immunofluorescence

Embryos were fixed in 4% paraformaldehyde, washed in PBS, dehydrated in 30% sucrose, and frozen in Optimal Cutting Temperature (OCT) medium. 10μm sections were rinsed in PBS, incubated 30 min at room temperature in blocking buffer (PBS + 0.2% Triton-X + 5% chicken and/or goat serum) and incubated overnight with the following primary antibodies: anti-GFP-Alexa 488 (ThermoFisher #A-21311; 1:100 dilution), anti-CD31 (BD Biosciences #550274; 1:200), anti-Meis1/2 (Santa Cruz #sc-10599; 1:200) or anti-Runx1 [EPR3099] (Abcam #ab92336; 1:200). Samples were washed three times with PBS, incubated one hour at room temperature with fluorochrome-conjugated secondary antibodies (1:200 dilution), further washed to remove unbound antibodies, and mounted in Prolong Gold mounting reagent with DAPI (ThermoFisher #P36941). For co-staining of Runx1 and GFP, samples were incubated with anti-GFP-Alexa 488 for 1 h only after washes of the secondary antibodies. Stained slides were scanned on an Axioplan II Zeiss microscope and chosen sections were imaged on a Zeiss AxioImager M2 microscope with ZEN 3.2 software or on a Nikon A1-si confocal.

### Gene expression analysis

Total RNA was extracted using Qiagen DNA/RNA AllPrep Micro kit and cDNA made with Invitrogen SuperScript II or Maxima H Minus reverse transcriptase. ddPCR was performed on a BioRad QX200 instrument using EvaGreen supermix (BioRad). Primers are listed in Supplementary Table 3.

### OP9 ex vivo assay

OP9 cells were plated in a four-well culture chamber slide (50,000 cells/well) or 96-well plate (5,000 cells/well) 24 h prior to the experiment. Cells from dissected AGM were directly sorted into each well and

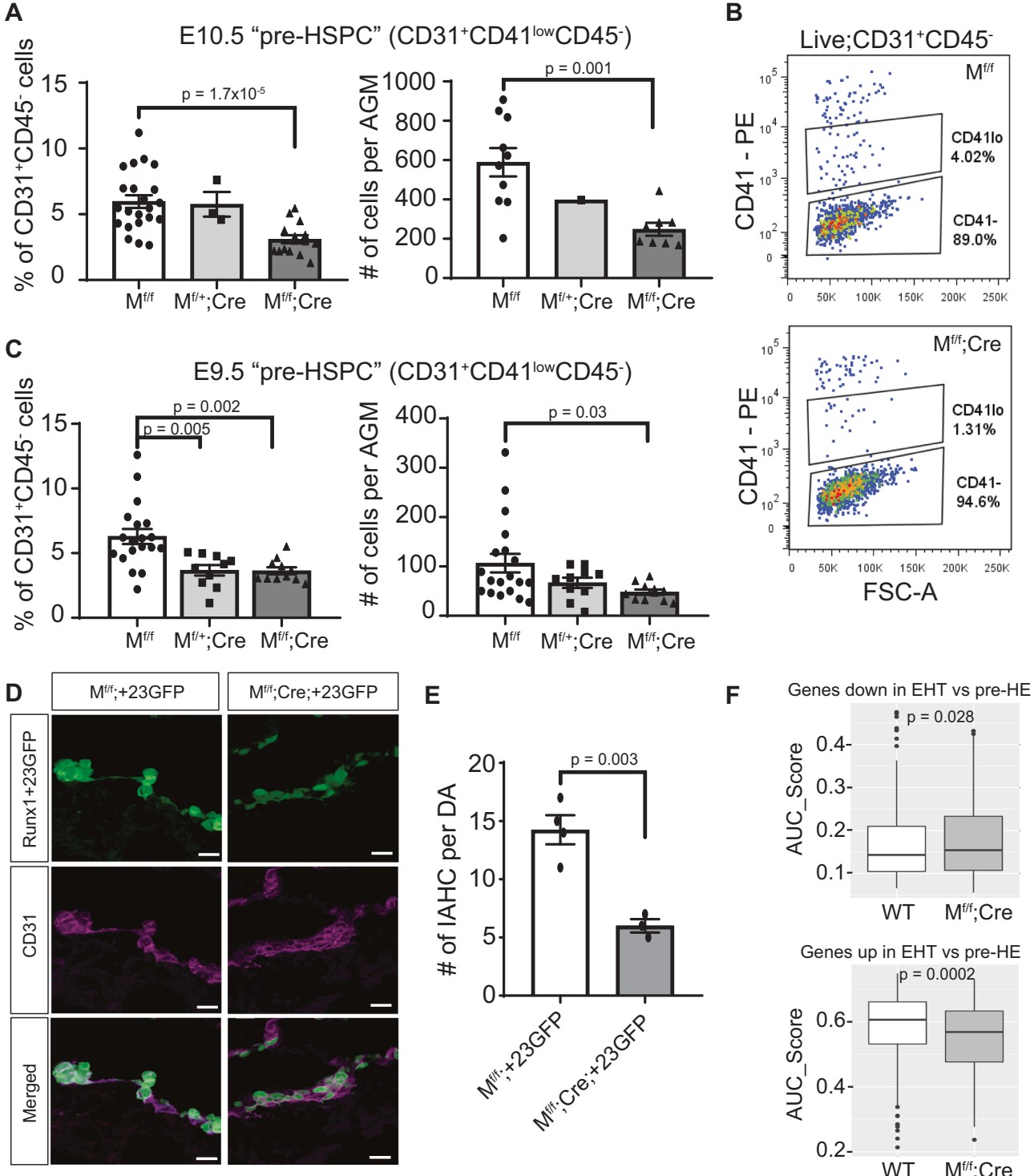

**Fig. 6 | Early defect in pre-HE significantly impairs the emergence of pre-HSPC.**
**A**, **C** Quantification of viable immunophenotypic pre-HSPC (CD31⁺CD41$^{low}$CD45⁻) by flow cytometry in (**A**) E10.5 embryos ($n = 22$ M$^{f/f}$, 3 M$^{f/+}$;Cre, and 15 M$^{f/f}$;Cre) and (**C**) E9.5 ($n = 19$ M$^{f/f}$, 10 M$^{f/+}$;Cre, and 11 M$^{f/f}$;Cre). In **A**, counting beads were used only on a subset of samples (right panel, $n = 10$ M$^{f/f}$, 1 M$^{f/+}$;Cre, and 8 M$^{f/f}$;Cre). Data are presented as mean ± SEM and each data point represents a different embryo.
**B** Representative flow cytometry scatterplots of CD41 expression on CD31⁺CD45⁻ cells at E10.5. **D** Immunostaining of E10.5 IAHC based on GFP (green) and CD31 (purple) expression in *Runx1*+23GFP embryos expressing *Meis1* (M$^{f/f}$;+23GFP) and littermates lacking endothelial expression of *Meis1* (M$^{f/f}$;Cre;+23GFP). Scale bar =

20 μm. **E** Quantification of large IAHC (>4 cells) per dorsal aorta ($n = 4$ M$^{f/f}$;+23GFP and 3 M$^{f/f}$;Cre;+23GFP embryos). Data are presented as mean ± SEM. **F** Gene expression scores for EHT gene signatures (downregulated (upper panel) or upregulated (lower panel)) in individual EHT cells from WT and M$^{f/f}$;Cre samples in the CITE-seq dataset, calculated using AUCell. Box-plots display the median (center line), the 25th and 75th percentiles (box limits), and ±1.5*interquartile range (whiskers). $n = 326$ WT cells from five independent samples and $n = 162$ M$^{f/f}$;Cre cells from three independent samples. All *p*-values in this figure were calculated using a two-sided t-test comparison between two groups. Source data are provided as a Source Data file.

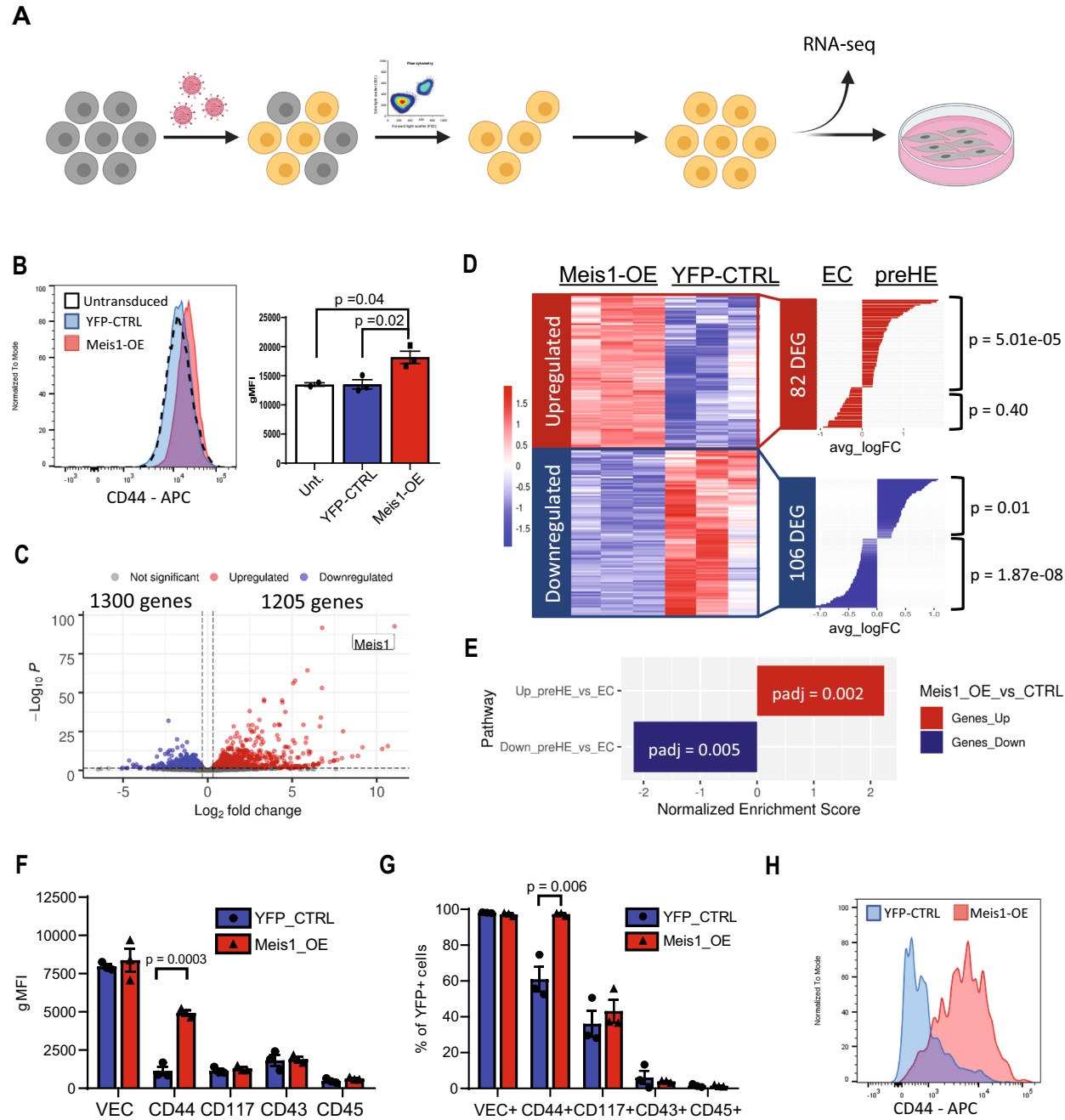

**Fig. 7 | Meis1 expression in EC triggers transcriptional changes akin to pre-HE cells. A** Schematic of the workflow used to investigate the effect of Meis1 over-expression in EC (created with BioRender.com). **B** Mean fluorescence intensity (MFI) of CD44 immunostaining 72 h post-transduction. Transduced EC were selected (VEC⁺CD31⁺YFP⁺) for the experiments (n = 2 parental untransduced CTRL, 3 YFP-CTRL, and 3 Meis1-OE). Bar plot on the right shows the mean ± SEM (p-values based on two-sided t-test). **C** Volcano plot showing DEG in Meis1-OE EC compared to YFP-CTRL EC by RNA-sequencing (p-values based on Wald test with Benjamini–Hochberg correction). **D** Heatmap depicting all genes up-/down-regu-lated in Meis1-OE EC in common with DEG in the CITE-seq analysis of aEC vs pre-HE cells. Hypergeometric test (one-tailed) was performed to determine whether the overlap was greater than expected from random sampling. **E** GSEA analysis quan-tifying enrichment of custom pre-HE genesets (based on CITE-seq expression) in Meis1-OE EC (p-values adjusted with Benjamini-Hochberg correction). **F**−**H** Transduced cells with Meis1-OE or YFP-CTRL vector were co-cultured on OP9 cells for 6 days followed by immunophenotyping. **F** gMFI values for the expression of surface markers relevant to EHT subpopulations and (**G**) percentage of cells expressing each marker. Data are presented as mean ± SEM and each data point depicts the average of technical duplicates (n = 3 independent biological samples for each group). P-values were calculated using a two-sided t-test. **H** Histogram plot of CD44 expression by flow cytometry after OP9 co-culture. Source data are pro-vided as a Source Data file.

incubated for 5 (E10.5) or 6 (E9.5) days in media supplemented with SCF (100 ng/ml), IL-3 (200 U/ml), G-CSF (100 ng/ml), Epo (2 U/ml), bFGF (1 ng/ml), and β-mercaptoethanol (50 μM). After 5 days of culture in chamber slide, cells were washed with PBS and stained with anti-mouse CD45 [30-F11] antibody (BD Biosciences #550539; 1:200 dilu-tion). For quantification of HE cells, AGM-derived EC were plated at the limiting dilution of 100 cells per well, a number that permits either one colony or no colonies to form in each well.

**Meis1-OE experiment**

In all, 100,000 primary EC were plated per well of a six-well plate precoated with Gelatin-Based Coating Solution (Cell Biologics) 24 h

prior transduction with Meis1-OE vector or YFP-CTRL (at an MOI of 20). Transduction was stopped after 48 h by complete media change and cells were FACS at 72 h to select for incorporation of the vector. Transduced EC (YFP⁺CD31⁺VEC⁺) were re-plated in a 6-well plate, expanded to confluency, and harvested. Total RNA was extracted from 300,000 cells using TRIzol reagent (Invitrogen) and between 300 and 400 ng of RNA was sent for ribodepleted RNA-seq at Canada's Michael Smith Genome Sciences Centre (Vancouver, Canada). Three independent replicates were performed for each condition. Differential gene expression analysis was performed using DESeq2[38] in R.

### CITE-seq

Embryos were dissected at E10.5 and AGM were processed for flow cytometry. Cells were stained with CD31-BV605 and the following CITE-seq antibodies at a final concentration of 0.2 μg/100 μL: Total-Seq™-A0012 anti-mouse CD117 (BioLegend #105843), TotalSeq™-A0443 anti-mouse CD41 (BioLegend #133937), TotalSeq™-A0110 anti-mouse CD43 (BioLegend #143211), TotalSeq™-A0157 anti-mouse CD45.2 (BioLegend #109853), TotalSeq™-A0191 anti-mouse/rat/human CD27 (BioLegend #124235), and TotalSeq™-A0073 anti-mouse/human CD44 (BioLegend #103045). Cells from 2 to 5 embryos were pooled for any given samples and CITE-seq was performed by the Biomedical Research Centre sequencing core at the University of British Columbia (Vancouver, British Columbia). Briefly, libraries were prepared using the 10X Chromium Single Cell 3′ Reagent v3 chemistry kit as per the standard protocol from 10X Genomics (Pleasanton, California) and sequenced on an Illumina Nextseq instrument.

### CITE-seq analysis

CITE-seq data was analyzed in R using the Seurat package[39]. First, cells were filtered based on the number of features detected (>2000 genes) and mitochondrial content (<10%). WT and M^{f/f};Cre samples were integrated using variance stabilizing transformation method (vst) based on the top 2000 variable features. Cells were then clustered based on transcriptomic gene expression using 50 principal components and the data represented using Uniform Manifold Approximation and Projection (UMAP). Each cluster was then manually assigned to a most likely cell type based on the top genes specifically enriched in that cluster as identified by non-parametric Wilcoxon rank sum test (FindMarker function in Seurat). For DEG between clusters, genes were considered differentially expressed at adjusted $p$-value < 0.1 based on Bonferroni correction.

Downstream analyses were performed as followed. Gene ontology (GO) term analysis was performed in R using gprofiler2[40]. Selected terms were reported as significantly enriched based on hypergeometric $p$-value < 0.05 after correction for multiple testing (using the default g:SCS algorithm). Gene expression scores from gene signature were calculated for individual cells in the CITE-seq data by using the AUCell package[41] in R. The arterial and venous scores were based on the expression of arterial (Dll4, Notch1, Notch4, Gja5, Nrp1, Jag1, Efnb2, Epas1, Vegfc) and venous (Nr2f2, Nrp2, Ephb4) genes[42]. scProportionTest (https://github.com/rpolicastro/scProportionTest)[43] was used to identify cluster over-/under-represented in M^{f/f};Cre samples compared to WT.

### RNA-sequencing

For transcriptome analysis of HE cells, CD31⁺CD117⁻CD45⁻ Runx1+23GFP⁻ cells were FACS-isolated from Runx1+23GFP transgenic embryos and compared to bulk EC FACS-isolated from WT littermates. Cells were lysed in Trizol and stored at −80 °C until library preparation and sequencing by Canada's Michael Smith Genome Sciences Centre (GSC) Biospecimen core. Samples were amplified using the SMART cDNA Amplification kit (Clontech, Mountain View, CA). Libraries were indexed and run in one lane of an Illumina Hiseq 2000 flow cell. The reads were aligned to Mus musculus genome mm9. Following sequencing, genes with a RPKM value ≥ 0.3 were compared between the two libraries and considered differentially expressed above a cut-off of 1.5-fold change. The list was filtered for TFs and input in the online tool DiRE[28]. The candidate TFs output from DiRE was further filtered to include only genes expressed in our HE dataset at E10.5 and the resulting hierachy was illustrated using Biotapestry[44].

### ChIP-seq

Epcr⁺CD150⁺CD48⁻lin⁻ cells were purified by FACS from adult C57BL/6 mice as previously described[45] and automatically deposited into viral-producer-coated 96-well plates. Cells were then transduced with Meis1-YFP and NA10HD-GFP as previously described[21]. Briefly, cells were infected by cocultivation with irradiated (4000 cGy) viral producers for NA10HD-GFP and Meis1-YFP in the presence of 5 μg/mL protamine sulfate (Sigma, Oakville, Canada) for 48 h. Cells were then flushed from the viral producers and grown for an additional 96 h in DMEM media, before being sorted by FACS for lin⁻GFP⁺Meis1-YFP⁺ transduced cells. At least $6 \times 10^6$ cells were isolated and chromatin immunoprecipitation (ChIP) was performed using an anti-HA antibody (Abcam cat# ab9110).

To identify Meis1 binding regions, sequence reads were aligned to the genome (mm9) using Burrows-Wheeler Alignment (BWA)[46] and peak enrichment was called using FindPeaks3. Genomic Regions Enrichment of Annotations Tool (GREAT)[47] was used with default settings (5 kb upstream, 1 kb downstream + 1000 kb extension) to find genes associated with binding peaks in the different datasets.

### Statistical analysis

Statistical analyses were calculated using two-tailed Student's $t$ test in GraphPad Prism 7.00 for Windows (GraphPad Software, La Jolla California USA) and considered to be statistically significant at $p < 0.05$. For all bar graphs with experimental replicates, individual data points are shown and the height of the bars represents the mean ± SEM (error bars). Experiment-specific statistical details can be found in respective figure legend. For experiments with multiple replicates, each measurement was taken from an independent sample.

### Reporting summary

Further information on research design is available in the Nature Portfolio Reporting Summary linked to this article.

## Data availability

The CITE-seq data, the RNA-seq data for HE cells, and the RNA-seq data for Meis1-OE cells have been deposited in the NCBI Gene Expression Omnibus under accession code GSE197244, GSE196047, and GSE197400, respectively. The ChIP-seq data have been deposited in the European Nucleotide Archive under accession code PRJEB52790. Co-expression data for the Meis1 analysis can be downloaded from the EnrichR webserver (https://maayanlab.cloud/Enrichr/). The published ChIP-seq datasets used in this study were downloaded from the original publications[30,31] Source data are provided with this paper.

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

## Acknowledgements

This work was supported by grants from the Canadian Institutes for Health Research (MOP-64354 and MOP-97744) to A.K.; P.C. was supported by a Canadian Institutes for Health Research studentship and a University of British Columbia doctoral fellowship; G.C. was supported by a Michael Smith Foundation for Health Research Trainee Award; A.K. is a Tier 1 Canada Research in Blood Cancers and is supported by the John Auston BC Cancer Foundation Clinical Investigator award.

## Author contributions

P.C. designed and performed research, analyzed and interpreted the data, and wrote the manuscript; G.C., A.F., J.D.K.P., and E.Y. performed experiments; L.D., P.L., and M.L. contributed to experiments; D.T. and M.B. assisted with data processing and analysis; R.K.H. provided Meis1 mouse strains and contributed to experimental design; A.K. conceived and designed the study, interpreted the data, and wrote the manuscript.

## Competing interests

The authors declare no competing interests.
