## [Peer Review File · Nature Communications]

Meis1 establishes the pre-hemogenic endothelial state prior to Runx1 expressionREVIEWER COMMENTS

Reviewer #1 (Remarks to the Author):

In this manuscript, the authors demonstrate that Meis1 is expressed at the earliest stages of EHT and distinguishes pre-HE cells primed towards the hemogenic trajectory from the arterial endothelial cells. Meis1 is functionally vital as endothelial deletion results in the impaired formation of functional Runx1-expressing HE, which significantly impedes the emergence of pre-HSPC via EHT. The authors, therefore, claim in their title that "Meis1 establishes the pre-hemogenic endothelial state prior to Runx1 expression".

The manuscript is well-structured and written, reports elegant experiments, and provides novel observations and findings. However, I have some comments and concerns, notably about interpreting some data.

Major comments:

1. There is a significant discrepancy between the frequency of Meis1 positive ECs detected through CITE-seq (a few %, fig 1A) and the GFP reporter (around 20-30%, fig 3C). The high frequency of Pre-HE/HE detected with the reporter is incompatible with the observation presented in the introduction that only 1.3% of EC isolated from E10.5 dorsal aorta have functional hematopoietic potential *ex vivo*. Therefore, it would be necessary to further validate the reporter by direct Meis1 immunostaining in this context. If the reporter is valid, this would indicate that the Meis1 expression detection by single-cell RNA-seq is suboptimal and that the frequency of EC expressing Meis1 is much higher than suggested. This finding would need to be further considered in the study.

2. Along this line, the authors should determine by ELDA the frequency of cells able to form hematopoietic colonies in the Meis1 ECs fraction. This approach would better quantify the enrichment/representation of functional preHE/HE in this fraction.

3. Ultimately, defining a strategy based on cell surface markers to isolate by flow cytometry preHE and HE specifically would be critical to evaluate further and contrast these two cell populations and definitively support some of the conclusions of this manuscript.

4. The preHE, and its transition to HE, is mainly defined here by the lack of RUNX1 detection by single-cell RNA-seq (see title), as in the precedent study from Zhu et al. However, a recent study indicates that a similar preHE population could be isolated with a RUNX1 reporter (Fadlullah et al., Blood, 2022). In this publication, the authors validated the correlation between the reporter and detection of RUNX1. As such, the absence of RUNX1 detection was shown to be the result of the poor sensitivity of the single-cell RNA-seq technique. This should be discussed in the manuscript and directly tested by evaluating the expression of RUNX1 in preHE cells by PCR or immunostaining (Meis1 positive ECs contain both preHE and HE cells). Alternatively, the Meis1 reporter could be combined with a RUNX1 reporter. In the absence of orthogonal experiments directly demonstrating the absence of RUNX1 in this preHE population, the title is probably not valid.

Minor comments.

1. The authors should give some indication about the performance of their single-cell RNAseq experiment, i.e., the depth of sequencing (median read numbers by cell), the number of genes detected per cell...

2. The RUNX1 immunostaining presented in figure 4E look suboptimal. More convincing immunostaining should be provided. RUNX1 should be detected in more cells, notably in the sub-aortic mesenchyme.

3. The authors indicate that they performed ELDA in figure 5B to quantify the frequency of HE in EC. In general, ELDA is based on serial dilutions of the cells to quantify the frequency of positive cells with high precision. In the assays presented, the authors seem to have averaged the frequency of preHE/HE in pools of 100 ECs from AGM from different embryos. This approach is less rigorous than doing a limiting dilution analysis; therefore, the term ELDA should be perhaps avoided.
4. The preHE is characterized in figure 1B by a striking higher expression of the endothelial markers Pecam, Cdh5, and Eng and the marker Dll4 than in normal ECs. How do the authors interpret this result?
5. The Meis1/GFP staining in the merged and magnified Meis1-GFP panels of the top of figure 4E seem different. Why?
6. Myeloid cell frequencies are increased in the absence of Meis1 (fig 5D/E). How do the authors interpret this surprising finding?

Reviewer #2 (Remarks to the Author):

Hematopoietic cells in the embryo form from a specialized population of endothelial cells called hemogenic endothelium. An immediate precursor of hemogenic endothelial cells has recently been described that is not well characterized. Coulombe et al. determine that the transcription factor Meis1 is expressed in the pre-HE population using single cell sequencing approaches. They further show using a combination of approaches including single cell sequencing and functional assays that Meis1 is required for the efficient generation of pre-HE cells from aortic endothelial cells. The work is interesting and useful. It corroborates an earlier description of a pre-HE population, and identifies a transcription factor important for its formation. A pre-HE population has also recently been described in human embryos, increasing the significance of this work. I have a few specific questions:

Figure 1A. There appears to be a branch of venous endothelial cells (vEC_4) leading to pre-HE in the UMAP. Can the authors comment on this? A trajectory analysis of the data would be useful for addressing the directionality and relationship of the vEC_4 population to pre-HE.

Figure 1A. At the end of the EHT trajectory are large populations on hematopoietic cells that are either EMPs or likely derived from EMPs. The authors don't really comment on this, and without explanation the trajectory makes it appear that the EMPs are differentiating from the arterial EHT population.

Figure 1F. The cells in the plot are labeled "EHT subset", but most of them are Runx1 and Gfi1 negative. Runx1 and Gfi1 define cells undergoing EHT, so the concept that only a subset of EHT cells are Runx1+ Gfi1+ HE cells is confusing.

Supplementary figure 1F. Should the label HE be EHT instead?

Figure 1 C,D. How do the pre-HE cells compare to the CD44 low Kit- population described by Oatley et al., and the pre-HE population described by Zhu et al.? It would be interesting to map the previously described cells on the authors' UMAP.

Figure 3G. The authors state that there is a relatively higher proportion of GFP+ cells at E9.5 than E10.5 in the EC compartment, but this not quantified in 3G. There should be a referral to a graph that shows this.

Pre-HSPC is a term that is not often used. Are these cells within intra-aortic hematopoietic clusters?
Are they Kit+?

Page 10, line 285. Figure 7C is mislabeled as 6C.

Did Meis1-overexpressing adult ECs upregulate Runx1 expression?

Reviewer #3 (Remarks to the Author):

Coulombe P. et al perform a detailed analysis of the AGM region using CITE-Seq technology. They identify a subset of cells that is characterized by the expression of Meis1 and based on dual expression of hemogenic and endothelial markers may define the branching point of these two lineages. Given the rarity of the identified subset and the absence of exclusive markers the authors here provide solid circumstantial evidence for the involvement of Meis1 as a key transcription factor for the initial specification of hematopoietic precursors from endothelial cells.

Specific comments:

Several trajectory algorithms have been developed, such as Monocle3, velocity,... to computationally infer developmental progression and lineage commitment. Given the limitations of these algorithms have you tried to apply any of those to your dataset?

Figure 4a is I believe mislabeled CD41 on the y axes should be CD44, please check.

In the ChIPSeq data please specific that the targets are defined based on proximity.

When you looked at the ChIP Seq data you just combined the list of targets with up-regulated genes, which is the most intuitive choice. However, negative regulated genes could play an essential role in lineage specification in particular considering that the Hemo-angiogenic split could require significant repression of the angiogenic related targets. It would be nice if you could look at that in parallel. (i.e. Nr2f2 looks as a specific genes that is downregulated (Fig1B))

When looking at the CITE-Seq data the Notch Ligand Dll4 is highly expressed on the identified HE-precursors. Following on your analysis, Meis1 expression anticipates the induction of Runx1, that is known to be a target of Notch1 in T cells. Further Notch-Dll4 signaling are acting on neighboring cells one expressing Notch receptors and one expressing Notch ligand. Have you considered the possibility that the expression of Dll4 is on cells that deliver the Notch signal that to the Receptor bearing cells that untimely become the committed HE-precursors? Have you looked on your datasets for Notch receptor expression?

Along these thoughts, we know that during T cell development a small fraction of cells receiving Notch signaling leaves the bone marrow and seeds the thymus to develop into T cells. Maybe a similar mechanism occurs here; where a minimal Notch signaling delivered through your pre-HE fraction of cells that expresses Dll4 leads to the Notch receptor expressing pre-HE to develop into the future fetal liver seeding hematopoietic progenitors.

Have you tried to culture your isolated fractions of Meis1-GFP- and Meis-1 GFP+ cells on OP9Delta? Please consider this more a curiosity rather than a revision requirement.

In several experiments it appears that CD44 expression (4a) segregates in high and low on Meis1+ cells. Based on the numbers is it possible to sort each fraction independently for analysis?

In Experiment of Fig 4. Cells were only sorted based on GFP+ for Meis1 expression. However, the

reasoning is that the committed precursor is the Meis1+ CD44-, would you have sufficient cells to perform bulk RNAseq or qPCR for validation of target genes?

To achieve targeted deletion of Meis1 you have used a pVE-Cre. Please highlight using the CiteSeq data of histology (3B) if an antibody is available the expected extent of Meis1 deletion.

to Fig. 6

The lack of bulging Runx1-Meis1 could this reflect a migration problem?

Have you looked in your data sets the chemokine receptor profiles? Is it possible that Meis1 the hemogenic program is impaired due to an inappropriate exposure to microenvironmental cues? Or alternatively, lack of proliferation may prevent the dilution of specific transcription factors? Have you checked for proliferation markers/stages across the generated data sets?

REVIEWER COMMENTS

Reviewer #1 (Remarks to the Author):

“In this manuscript, the authors demonstrate that Meis1 is expressed at the earliest stages of EHT and distinguishes pre-HE cells primed towards the hemogenic trajectory from the arterial endothelial cells. Meis1 is functionally vital as endothelial deletion results in the impaired formation of functional Runx1-expressing HE, which significantly impedes the emergence of pre-HSPC via EHT. The authors, therefore, claim in their title that “Meis1 establishes the pre-hemogenic endothelial state prior to Runx1 expression”.

The manuscript is well-structured and written, reports elegant experiments, and provides novel observations and findings. However, I have some comments and concerns, notably about interpreting some data.

Major comments:

1. There is a significant discrepancy between the frequency of Meis1 positive ECs detected through CITE-seq (a few %, fig 1A) and the GFP reporter (around 20-30%, fig 3C). The high frequency of Pre-HE/HE detected with the reporter is incompatible with the observation presented in the introduction that only 1.3% of EC isolated from E10.5 dorsal aorta have functional hematopoietic potential ex vivo. Therefore, it would be necessary to further validate the reporter by direct Meis1 immunostaining in this context. If the reporter is valid, this would indicate that the Meis1 expression detection by single-cell RNA-seq is suboptimal and that the frequency of EC expressing Meis1 is much higher than suggested. This finding would need to be further considered in the study.”

We understand that the reviewer is concerned about the detection of Meis1 in some experiments. With respect to the *Meis1*-GFP reporter mice, we have previously shown that GFP correlates very well with Meis1 protein level in these mice [1]. In this paper, we have shown that only *Meis1*-GFP⁺ cells express Meis1 transcripts at the RNA level (Fig. 4B). In addition, we have now further validated that GFP expression in the AGM of *Meis1*-GFP embryos matches anti-Meis1/2 immunostaining and have added a Supplementary Figure to highlight the concordance (Suppl. Fig. 5A). The 21.8% GFP positive cells gated by FACS (mean at E10.5 in Fig. 3C) represent a heterogeneous population of EC, pre-HE, and HE cells, the latter two clearly expressing high levels of Meis1. This proportion is comparable with what has previously been reported using cell surface markers (CD44⁺ cells constitute roughly 30% of all VE-Cadherin positive cells at E10.5 [2] and ACE⁺ cells represent 27% of EC [3]). Therefore, we are confident of the validity of this reporter mouse model.

We agree with the reviewer that there appears to be a discrepancy in the scRNA-seq data at first glance, but perhaps not as significant as it seems. In fact, if we look very carefully at the data, we can find non-zero counts for Meis1 in 9.9% of all the EC populations in the scRNA-seq dataset. With the addition of pre-HE and HE/EHT cells in the scRNA-seq (roughly 2.6% and 5% respectively of the total EC compartment), we achieve a percentage (14.8% cells with non-zero

counts) that is comparable to the FACS data (where GFP is a proxy for Meis1 protein level). However, Meis1 expression in EC is very low and the difference between Meis1 expression in pre-HE and EC in the scRNA-seq remains significant. Immunostaining of the AGM reveals additional GFP⁺ cells adjacent to and surrounding the emerging IAHC clusters (Fig. 3B, 4E). While we have not explored this phenomenon extensively, it is possible that some *Meis1*⁺ cells are primed to undergo EHT, but eventually get repressed and retain EC signatures as a neighbouring cell enters the transdifferentiation process. There are likely other factors that act in concert with Meis1 in order for EHT to initiate, highlighting a potential influence of the microenvironment of each individual EC and the relevant complement of expressed RNAs/proteins. While this would be an interesting direction to pursue, we feel that further evaluation of these details is beyond the scope of the current manuscript.

“2. Along this line, the authors should determine by ELDA the frequency of cells able to form hematopoietic colonies in the Meis1 ECs fraction. This approach would better quantify the enrichment/representation of functional preHE/HE in this fraction.”

We agree with the reviewer that this quantification is important, and in fact we had performed a limiting dilution assay to show that only Meis1 ECs can form hematopoietic colonies (Fig. 4F). We have now added the ELDA frequency data to this figure. This value (1/270 Meis1⁺ EC represent functional pre-HE/HE) is very close to the frequency reported by Zhu et al. in CD44⁺ cells (1/276; [4]) which would be expected given the overlap of expression observed between Meis1 and CD44 at E10.5 (Fig. 3).

“3. Ultimately, defining a strategy based on cell surface markers to isolate by flow cytometry preHE and HE specifically would be critical to evaluate further and contrast these two cell populations and definitively support some of the conclusions of this manuscript.”

While we certainly agree with the reviewer that defining a strategy to isolate pre-HE and HE cell by flow cytometry using only surface markers would ultimately be beneficial, the main objective of this manuscript was to identify transcriptional programs/regulators that promote EHT emergence. To this end, we identified Meis1 as an important transcription factor, characterized its expression based on known surface markers, showed that Meis1 is expressed in all Runx1⁺ HE but also in some adjacent Runx1⁻ cells, and validated the functional significance of Meis1. As mentioned in the discussion, the *Meis1*-GFP mice (which will soon become available from JAX) could be crossed with other genetically perturbed mice to better define pre-HE and HE cells, although this is admittedly not as convenient as using surface markers.

Thus, in order to address the Reviewer's comment, we identified surface markers that are expressed at the transcriptomic level in pre-HE cells but absent in the EHT cluster using both our CITE-seq dataset and that of Zhu et al [4]. Given that most candidate surface markers were also expressed to some level in EC, we combined these markers with CD44 to distinguish between EC (CD44⁻), pre-HE (CD44⁺marker⁺) and HE (CD44⁺marker⁻). We selected 2 markers from the list with commercially available antibodies (CD200 and Notch4) to test further. For both these markers, we FACS-isolated EC, pre-HE, HE, and pre-HSPC, as per the schematic below, from

pools of E10.5 embryos and examine RNA expression of *Meis1*, *Runx1*, *Spi1*, and selected *Meis1* target genes enriched in pre-HE cells by ddPCR. We find that the CD44⁺CD200⁺ (or Notch4⁺) fraction expressed high levels of pre-HE genes (*Vegfc*, *Tulp4*, *Smad6*; see Supplementary Fig. 5C and 7D-E), but did not express *Runx1* or *Spi1* which is upregulated in CD44⁺CD200⁻ (or Notch4⁻) cells. Based on our definition of pre-HE and HE cells from the gene signatures in the CITE-seq data, we can thus isolate CD44⁺CD200⁺ (or Notch4⁺) pre-HE and CD44⁺CD200⁻ (or Notch4⁻) HE cells by flow cytometry. All data is shown in the new Supplementary Fig. 7. A paragraph was added on page 7-8 of the manuscript to describe these new findings.

“4. The preHE, and its transition to HE, is mainly defined here by the lack of *RUNX1* detection by single-cell RNA-seq (see title), as in the precedent study from Zhu et al. However, a recent study indicates that a similar preHE population could be isolated with a *RUNX1* reporter (Fadlullah et al., Blood, 2022). In this publication, the authors validated the correlation between the reporter and detection of *RUNX1*. As such, the absence of *RUNX1* detection was shown to be the result of the poor sensitivity of the single-cell RNA-seq technique. This should be discussed in the manuscript and directly tested by evaluating the expression of *RUNX1* in preHE cells by PCR or immunostaining (*Meis1* positive ECs contain both preHE and HE cells). Alternatively, the *Meis1* reporter could be combined with a *RUNX1* reporter. In the absence of orthogonal experiments directly demonstrating the absence of *RUNX1* in this preHE population, the title is probably not valid.”

We appreciate the Reviewer’s concerns here, and had similar thoughts when performing these investigations. Thus, in our original submission, we performed orthogonal experiments in this manuscript to show that *Meis1* is expressed in *Runx1*-negative cells while the opposite is not true. Evidence to support this statement can be seen at the protein level in the immunostaining of Fig. 4E. Similarly, we also examined *Runx1* expression in *Meis1*⁺ EC by ddPCR in Fig. 4B. We detected *Runx1* expression only in the *Meis1*⁺ population, which contains HE cells (as mentioned by the Reviewer), but not in the *Meis1*⁻ population. In contrast, *Meis1* expression was detected in

both the Runx1⁺ and the Runx1⁻ cells (Fig. 4C). These findings indicate the presence of a population that is *Meis1*⁺ and *Runx1*⁻, consistent with *Meis1* expression preceding *Runx1*. In addition, in the population sorted based on the novel surface markers identified (see response to comment 3 above) we also observed stronger expression of *Meis1* in the more primitive subset. *Meis1* expression is relatively constant across these subsets (CD44⁺CD200⁺ pre-HE, CD44⁺CD200⁻ HE and CD41^{low} pre-HSPC), whereas *Runx1* expression increases significantly. Thus, all our observations point to a population of cells expressing *Meis1*, but low/undetectable levels of *Runx1*.

Minor comments.

“1. The authors should give some indication about the performance of their single-cell RNAseq experiment, i.e., the depth of sequencing (median read numbers by cell), the number of genes detected per cell...”

We thank the reviewer for this suggestion. We have now added these scRNA-seq metrics to a new Supplementary Table 1.

“2. The RUNX1 immunostaining presented in figure 4E look suboptimal. More convincing immunostaining should be provided. RUNX1 should be detected in more cells, notably in the sub-aortic mesenchyme.”

We thank the Reviewer for noticing this. We believe that the detection of *Runx1* works well, and we have provided additional immunostaining (see new Supplementary Fig. 6) to support this point. As mentioned by the reviewer, *Runx1* is present and detected in the sub-aortic mesenchyme. The amount of *Runx1* that we observe varies between tissue sections examined. We had previously chosen to show areas where IAHC were clearly present in the dorsal aorta (over sections of *Runx1* in the mesenchyme), but the new Supplementary Fig. 6 is consistent with the Reviewer's point. In addition, we validated the immunostaining with a *Runx1*+23GFP reporter mouse (also shown in the new Supplementary Fig. 6). The *Runx1* antibody used for these experiments is the same antibody that was used by Fadlullah et al, Blood 2022 (referenced above by the Reviewer) and has been validated.

“3. The authors indicate that they performed ELDA in figure 5B to quantify the frequency of HE in EC. In general, ELDA is based on serial dilutions of the cells to quantify the frequency of positive cells with high precision. In the assays presented, the authors seem to have averaged the frequency of preHE/HE in pools of 100 ECs from AGM from different embryos. This approach is less rigorous than doing a limiting dilution analysis; therefore, the term ELDA should be perhaps avoided.”

In figure 5B, each data point on the plot represents the outcome of a single embryo, where the frequency was determined by (# of wells with colony / # cells plated). The average for each genotype was then estimated using ELDA statistics at one limiting dilution, where the dose was 100 cells (based on the limiting dilution assay in Figure 4F), the number of tests correspond to

the total number of wells plated, and the response was the total number of wells with colonies across all replicates. We have now clarified this in the Figure legend, and replaced ELDA frequency with HE frequency. However, we note that a true limiting dilution assay was reported in Figure 4F to determine the frequency of HE in Meis1⁺ and Meis1⁻ EC.

*“4. The preHE is characterized in figure 1B by a striking higher expression of the endothelial markers *Pecam*, *Cdh5*, and *Eng* and the marker *Dll4* than in normal ECs. How do the authors interpret this result?”*

As noted by the reviewer, we see high expression of endothelial genes and genes related to angiogenesis in our pre-HE population. We hypothesize that this could be due to the “migratory” nature of these cells. While we can only speculate at this point, we know that during angiogenesis the tip cells at the leading edge of the sprout express higher levels of Dll4 for instance. Whether a similar mechanism is at play during EHT initiation, as pre-HE reorganize their structure to reach into the vascular lumen, would require further investigation.

*“5. The *Meis1*/GFP staining in the merged and magnified *Meis1*-GFP panels of the top of figure 4E seem different. Why?”*

The magnified panel is a different higher power image limited to the boxed region that was taken using a higher objective on the microscope and is not a computer-based zoomed-in version of the merged panels. It is possible that the plane and position of the image differs slightly.

*“6. Myeloid cell frequencies are increased in the absence of *Meis1* (fig 5D/E). How do the authors interpret this surprising finding?”*

This is an interesting question. First, we believe that the cells with a myeloid signature are blood cells circulating in the dorsal aorta at the time of dissection. Given that they are circulating cells, one possibility could simply be that there is variability between samples in the number of cells being “flushed out” of the dorsal aorta during the dissection. However, we did not observe significant differences in the percentage of CD45⁺ hematopoietic cells by flow cytometry (see below, left panel). Our scRNA-seq dataset is based only on CD31⁺ cells. Therefore, when we looked at co-expression of CD45 within the CD31⁺ compartment by flow cytometry, we see a similar trend as in the scRNA-seq data (see below, right panel). We know that in the adult mouse bone marrow, Meis1 is required for erythropoiesis and megakaryopoiesis [5]. Therefore, one hypothesis could be that the loss of Meis1 skews primitive hematopoiesis towards a myeloid state or a more immature population. Interestingly, *Vegfc*, a predicted Meis1 target in our dataset and a gene enriched in pre-HE cells, was reported to play a role in fate determination. Specifically, loss of *Vegfc* in zebrafish embryos results in decreased numbers of HSPCs and an abundance of myeloid progenitors [6]. Therefore, Meis1-deletion might skew hematopoiesis towards myeloid differentiation via decreased *Vegfc* expression. However, we feel that defining the reason for this change is beyond the scope of the current manuscript.

“Reviewer #2 (Remarks to the Author):

Hematopoietic cells in the embryo form from a specialized population of endothelial cells called hemogenic endothelium. An immediate precursor of hemogenic endothelial cells has recently been described that is not well characterized. Coulombe et al. determine that the transcription factor Meis1 is expressed in the pre-HE population using single cell sequencing approaches. They further show using a combination of approaches including single cell sequencing and functional assays that Meis1 is required for the efficient generation of pre-HE cells from aortic endothelial cells. The work is interesting and useful. It corroborates an earlier description of a pre-HE population, and identifies a transcription factor important for its formation. A pre-HE population has also recently been described in human embryos, increasing the significance of this work. I have a few specific questions:

Figure 1A. There appears to be a branch of venous endothelial cells (vEC_4) leading to pre-HE in the UMAP. Can the authors comment on this? A trajectory analysis of the data would be useful for addressing the directionality and relationship of the vEC_4 population to pre-HE.”

We thank the Reviewer for this suggestion. Indeed, the vEC_4 cells are proximal to the pre-HE population in the UMAP projection. Interestingly, when we performed *in silico* trajectory analysis using Monocle3, the branching to pre-HE and EHT cells seems to emerge directly and exclusively from the aEC population. We added this trajectory analysis in Supplementary Fig. 1E. This observation is consistent with the correlation that we observed between the “conflux endo” population in Zhu et al. [4] that is thought to give rise to preHE cells and our “aEC” population (see also response to comment below regarding cell mapping between datasets)

“Figure 1A. At the end of the EHT trajectory are large populations on hematopoietic cells that are either EMPs or likely derived from EMPs. The authors don’t really comment on this, and without explanation the trajectory makes it appear that the EMPs are differentiating from the arterial EHT population.”

We thank the reviewer for pointing out that we failed to discuss the EMPs in the original version of the manuscript. These cells could be IAHC cells that have acquired hematopoietic gene expression signatures but they are likely derived from the yolk sac. We added a sentence (page 3, line 11) to reflect this point.

“Figure 1F. The cells in the plot are labeled “EHT subset”, but most of them are Runx1 and Gfi1 negative. Runx1 and Gfi1 define cells undergoing EHT, so the concept that only a subset of EHT cells are Runx1+ Gfi1+ HE cells is confusing.”

We apologize for the confusion here. The plot in Fig. 1F shows only the cells from the “EHT subset” (labeled above the plot) in the UMAP in Fig.1A. These cells are all Runx1⁺ and most are Gfi1⁺ at the transcriptomic level (towards the more “mature” end of the cluster, some cells have downregulated Gfi1 and upregulated Gfi1b as we would expect; see Fig. 4D and below). The intent of Fig.1F was to show the presence of “HE” cells within that “EHT subset”. These “HE” cells are Runx1⁺ and Gfi1⁺ at the transcriptomic level but surface c-Kit⁻ and CD41⁻ based on ADT (protein level). The rest of the “EHT” cells express surface markers associated with EHT progression such as c-Kit, CD41, and CD43. We modified the text to make that point clearer (page 4, lines 3-4).

“Supplementary figure 1F. Should the label HE be EHT instead?”

The original Supplementary Fig. 1F (now Supplementary Fig. 1I) shows the lack of surface marker expression (CD117, CD41, CD43, and CD45) on HE cells as compared to pre-HSPC cells undergoing EHT. Thus, we have left the label as is and added a sentence to the figure legend.

“Figure 1 C,D. How do the pre-HE cells compare to the CD44 low Kit- population described by Oatley et al., and the pre-HE population described by Zhu et al.? It would be interesting to map the previously described cells on the authors’ UMAP.”

We thank the reviewer for this interesting suggestion. Mapping the cell populations from Zhu et al. onto our UMAP shows an excellent correlation of the populations. Specifically, 91.2% of their pre-HE cells map to our pre-HE population. Zhu et al. used different cell sorting strategies and have more “confluent AE” leading to pre-HE. We added this UMAP as a new panel in suppl Fig. 1C-D. We tried mapping the CD44^{low} c-Kit⁻ from Oatley et al. to our dataset. However, the

mapping was not robust given that they used single-cell q-RT-PCR on a narrow list of 96 pre-selected genes. Their CD44^{low}Kit⁻ cells map to our pre-HE/pre-HE (58%) and EHT (42%) populations but we know from our CITE-seq data that our EHT cells are c-Kit⁺. Therefore, this is not as accurate as the comparison with other scRNA-seq datasets containing >2000 expressed features per cells.

“Figure 3G. The authors state that there is a relatively higher proportion of GFP⁺ cells at E9.5 than E10.5 in the EC compartment, but this not quantified in 3G. There should be a referral to a graph that shows this.”

The Reviewer is right that Fig. 3G does not show quantification. However, the graph in Fig. 3C shows a higher proportion of GFP⁺ cells at E9.5 in the EC compartment vs E10.5. We added a proper referral to this figure after the statement on page 6, line 24: “Tracking *Meis1* expression at an earlier time point showed a relatively higher proportion of GFP⁺ cells at E9.5 compared to E10.5 specifically within the EC compartment (Fig. 3C, G)”

“Pre-HSPC is a term that is not often used. Are these cells within intra-aortic hematopoietic clusters? Are they Kit⁺?”

The term frequently found in the literature is pre-HSC. However, there is no evidence that all “pre-HSC” become a functional HSC and there is growing evidence that EHT can produce other progenitors/blood cells without going through the “HSC” state [7]. As such, we feel that the term pre-HSPC reflects more accurately the potential of these cells. These are c-Kit⁺ cells found within the intra-aortic hematopoietic clusters.

“Page 10, line 285. Figure 7C is mislabeled as 6C.”

We thank the reviewer for catching this typo and we have corrected the labeling in the revised version of the manuscript.

*“Did *Meis1*-overexpressing adult ECs upregulate *Runx1* expression?”*

We found that overexpressing *Meis1* in adult ECs upregulated many relevant genes but *Runx1* was not one of them. This is very likely contributing to the fact that these *Meis1*-OE ECs do not undergo complete EHT under these culture conditions. While *Meis1* is required for initiation of EHT, it does not appear sufficient to complete the process, at least in the context of adult aortic EC under the culture conditions used.

“Reviewer #3 (Remarks to the Author):

*Coulombe P. et al perform a detailed analysis of the AGM region using CITE-Seq technology. They identify a subset of cells that is characterized by the expression of *Meis1* and based on dual*

expression of hemogenic and endothelial markers may define the branching point of these two lineages. Given the rarity of the identified subset and the absence of exclusive markers the authors here provide solid circumstantial evidence for the involvement of Meis1 as a key transcription factor for the initial specification of hematopoietic precursors from endothelial cells.

Specific comments:

Several trajectory algorithms have been developed, such as Monocle3, velocity,... to computationally infer developmental progression and lineage commitment. Given the limitations of these algorithms have you tried to apply any of those to your dataset?"

We thank the Reviewer for this suggestion. We performed *in silico* trajectory analysis using Monocle3 that we have now included in a new panel of Supplementary Fig. 1E. This analysis reveals several branching trajectories within the various EC subsets but most interestingly, the trajectory to pre-HE and EHT cells emerge directly from the aEC population.

"Figure 4a is I believe mislabeled CD41 on the y axes should be CD44, please check."

We apologize for the confusion. The axis labeling in Fig. 4A is accurate but the cell population labeled on top of the plot was mislabeled. The cells shown in Fig. 4A are from a "Live; CD31⁺CD45⁻" gate and in this flow plot, the y-axis should be CD41 as stated. The cell population labeled is corrected in the revised version of this figure.

"In the ChIPSeq data please specific that the targets are defined based on proximity."

We thank the Reviewer for raising this point. We included the following statement in the Results section of the revised manuscript (page 5, line 23): "mapping to 16,198 nearby genes based on proximity"

"When you looked at the ChIP Seq data you just combined the list of targets with up-regulated genes, which is the most intuitive choice. However, negative regulated genes could play an essential role in lineage specification in particular considering that the Hemo-angiogenic split could require significant repression of the angiogenic related targets. It would be nice if you could look at that in parallel. (i.e. Nr2f2 looks as a specific genes that is downregulated (Fig1B))"

We thank the Reviewer for this important suggestion. We looked in parallel at potential regulation that would lead to downregulation of gene expression. The resulting GO terms are quite broad (ie: "regulation of cellular process" and "regulation of biological process" as top hits). Looking at the list of genes more closely, we find interesting candidates that are potentially repressed by Meis1 including angiogenic related targets Nr2f2 (as the reviewer rightfully points out) and Nrp2, in addition to Wnt-related targets such as Foxq1 and Nkd1 that also need to be repressed. We added this figure in a new Supplementary Fig. 4, in addition to a statement in the manuscript (page 6, line 12-16)

“When looking at the CITE-Seq data the Notch Ligand Dll4 is highly expressed on the identified HE-precursors. Following on your analysis, Meis1 expression anticipates the induction of Runx1, that is known to be a target of Notch1 in T cells. Further Notch-Dll4 signaling are acting on neighboring cells one expressing Notch receptors and one expressing Notch ligand. Have you considered the possibility that the expression of Dll4 is on cells that deliver the Notch signal that to the Receptor bearing cells that untimely become the committed HE-precursors? Have you looked on your datasets for Notch receptor expression?”

Along these thoughts, we know that during T cell development a small fraction of cells receiving Notch signaling leaves the bone marrow and seeds the thymus to develop into T cells. Maybe a similar mechanism occurs here; where a minimal Notch signaling delivered through your pre-HE fraction of cells that expresses Dll4 leads to the Notch receptor expressing pre-HE to develop into the future fetal liver seeding hematopoietic progenitors.

Have you tried to culture your isolated fractions of Meis1-GFP- and Meis-1 GFP+ cells on OP9Delta? Please consider this more a curiosity rather than a revision requirement.”

We thank the Reviewer for this stimulating input. Clearly, the Notch pathway and Dll4 play critical roles during EHT. We find that Notch1 is widely expressed across our EC population and highly expressed in pre-HE cells at the transcriptomic level. The study of the Notch pathway in this context has been the subject of many publications, including many reviews, and we didn't explore this pathway further within the scope of this manuscript. An elegant report recently showed that Dll4 expression controls the size of the clusters, by limiting the recruitment of adjacent HE cells [8]. In that context, Dll4 appears to be expressed on the IAHC cells and sends an inhibitory signal to neighboring cells. Based on the literature [8, 9], we believe that Dll4 expression on pre-HE might limit the acquisition of a hemogenic program in adjacent cells by reinforcing an endothelial fate. Other Notch ligands are also involved in balancing Notch signal strength, namely Jag1 which, in contrast to Dll4, seems to promote the hemogenic transition in receiving cells [9].

We only cultured cells on parental OP9 cells as it is common practice in the context of EHT and have not tried using OP9-Dll4 to grow our Meis1^{+/-} EC. Based on previous reports, OP9-Dll4 cells are not optimal to derive CD45⁺ cells in this particular context [9].

“In several experiments it appears that CD44 expression (4a) segregates in high and low on Meis1+ cells. Based on the numbers is it possible to sort each fraction independently for analysis?”

In Experiment of Fig 4. Cells were only sorted based on GFP+ for Meis1 expression. However, the reasoning is that the committed precursor is the Meis1+ CD44-, would you have sufficient cells to perform bulk RNAseq or qPCR for validation of target genes?”

The reviewer seems to be referring to Fig. 4a here, which shows CD41 expression (and not CD44 as explained in a previous comment). CD41 expression can be segregated between high and mid, with the CD41^{mid} population containing pre-HSC [10]. With regards to CD44, our staining shows mostly two distinct populations of CD44⁺ and CD44⁻ within the GFP⁺ EC.

Following the Reviewer's suggestion in the second part of the comment, we were able to sort enough GFP⁺CD44⁻ and GFP⁺CD44⁺ cells at E9.5 (a timepoint where there is a better representation of both subsets) from a total of 28 embryos pooled in 3 groups, to look at gene expression by ddPCR. We looked at selected genes that are considered Meis1 targets based on the ChIP-seq (common to all three datasets) and upregulated in the over-expression experiment. The selected genes also had a relatively high expression in pre-HE compared to aEC and EHT cells (Supplementary Fig. 5C). The result of this experiment (shown in Supplementary Fig. 5) suggests that the GFP⁺CD44⁺ gate contains HE cells (Runx1 and Spi1 expression) while GFP⁺CD44⁻ cells have relatively higher levels of the selected pre-HE/Meis1 target genes. Although pre-HE cells express CD44 at E10.5 (based on CITE-seq data, Fig. 1C), this experiment shows that E9.5 GFP⁺CD44⁻ cells already appear primed towards a pre-HE expression profile (Supplementary Fig. 5).

Gating for sort:

“To achieve targeted deletion of Meis1 you have used a pVE-Cre. Please highlight using the CiteSeq data of histology (3B) if an antibody is available the expected extent of Meis1 deletion.”

We believe that the Reviewer is asking here to validate the change in Meis1 protein level in our VE-Cre model. We measured the extent of Meis1 deletion by ddPCR at the genomic level (Supplementary Fig. 5) and transcript level (Fig. 5C) to support the knockdown efficiency in that model. In addition, we have now further validated Meis1 knockdown at the protein level via immunostaining of Meis1 in E10.5 Meis1-flox VEC WT and KO embryos. Antibody staining confirms a lack of Meis1 protein expression in the endothelium and this data has been added to supplementary Fig. 9D.

“to Fig. 6

The lack of bulging Runx1-Meis1 could this reflect a migration problem?

Have you looked in your data sets the chemokine receptor profiles? Is it possible that Meis1 the

hemogenic program is impaired due to an inappropriate exposure to microenvironmental cues? Or alternatively, lack of proliferation may prevent the dilution of specific transcription factors? Have you checked for proliferation markers/stages across the generated data sets?"

This is an interesting observation from the Reviewer. While we cannot fully rule out possible microenvironmental cues, we believe that Meis1 is acting in a cell-autonomous fashion. In particular, *ex vivo* microenvironmental cues did not rescue the defect in functional hemogenic potential in Meis1-deficient cells (Fig. 5B). Regarding the chemokine receptor profiles, we looked at the geneset from the GO term “chemokine receptor activity” as a whole but observed a low detection of transcripts in the pre-HE cells of our dataset. Therefore, we are limited in our ability to accurately quantify differences between WT and Meis1-null cells for this set of genes (panel A below). However, we see that proliferation does not significantly differ between the two genotypes based on cell cycle scoring analysis (Seurat package) as shown below (panel B).

References

1. Xiang, P., et al., *A knock-in mouse strain facilitates dynamic tracking and enrichment of MEIS1*. Blood Adv, 2017. **1**(24): p. 2225-2235.
2. Oatley, M., et al., *Single-cell transcriptomics identifies CD44 as a marker and regulator of endothelial to haematopoietic transition*. Nature communications, 2020. **11**(1): p. 586-586.
3. Fadlullah, M.Z.H., et al., *Murine AGM single-cell profiling identifies a continuum of hemogenic endothelium differentiation marked by ACE*. Blood, 2022. **139**(3): p. 343-356.
4. Zhu, Q., et al., *Developmental trajectory of prehematopoietic stem cell formation from endothelium*. Blood, 2020. **136**(7): p. 845-856.
5. Miller, M.E., et al., *Meis1 Is Required for Adult Mouse Erythropoiesis, Megakaryopoiesis and Hematopoietic Stem Cell Expansion*. PLoS One, 2016. **11**(3): p. e0151584.
6. Schiavo, R.K. and O.J. Tamplin, *Vascular endothelial growth factor c regulates hematopoietic stem cell fate in the dorsal aorta*. Development, 2022. **149**(2).
7. Dignum, T., et al., *Multipotent progenitors and hematopoietic stem cells arise independently from hemogenic endothelium in the mouse embryo*. Cell Rep, 2021. **36**(11): p. 109675.
8. Porcheri, C., et al., *Notch ligand Dll4 impairs cell recruitment to aortic clusters and limits blood stem cell generation*. The EMBO Journal, 2020. **39**(8): p. e104270.
9. Gama-Norton, L., et al., *Notch signal strength controls cell fate in the haemogenic endothelium*. Nature Communications, 2015. **6**: p. 8510.

10. Rybtsov, S., et al., *Hierarchical organization and early hematopoietic specification of the developing HSC lineage in the AGM region*. *The Journal of Experimental Medicine*, 2011. **208**(6): p. 1305-1315.

REVIEWERS' COMMENTS

Reviewer #1 (Remarks to the Author):

The authors have addressed most of my concerns and comments. I have a few remaining questions that need to be answered.

1. When introducing the CITE-seq experiment, it would be informative to include directly in the text (first sentence of results) the antibodies used on CD31+ cells and the rationale for this choice. I understand they are CD44, CD117, CD41, CD43, CD45.2, and CD27.
2. Avoid/eliminate the use of primitive in the sentence "These mature hematopoietic cells, including EMPs, are likely derived from primitive hematopoiesis in the yolk sac but may also represent a few mature cells generated at this stage from the EHT process" as primitive is often used to describe the first wave of primitive erythroid cells that are not giving rise to EMPs.
3. The observation that many endothelial cells express Runx1+23GFP in the absence of Meis1 (Fig6D) is intriguing and interesting. These results suggest that Meis1 is required for EHT instead of preHE specification as proposed (as no RUNX1 expression would be expected in this later case). Could the authors comment on this observation? Could the authors look for Runx1, Gfi1 expression in their scRNAseq of the Meis1 KO (Fig5D)?
4. It is still incorrect to show in Fig4E some images that do not correspond directly to the amplification of the boxed area, as this would be the expectation of the reader. The data should be presented differently to avoid this ambiguity.
5. The authors addressed my comment "The RUNX1 immunostaining presented in figure 4E looks suboptimal. More convincing immunostaining should be provided. RUNX1 should be detected in more cells, notably in the sub-aortic mesenchyme." by providing in the revision, new Runx1 staining in figure S6A. However, I was not able to understand this figure S6A in the absence of an annotation or appropriate figure legend. Also, it needs to be clarified how providing a RUNX1 staining alone will address the potential problem with the Meis1/RUNX1/CD31 staining presented in Fig 4E. The staining in Fig 4E should be repeated.
6. The authors addressed my concern about a significant discrepancy between the frequency of Meis1 positive ECs detected through CITE-seq (a few %, fig 1A) and the GFP reporter (around 20-30%, fig 3C) by indicating the presence of non-zero counts for Meis1 in 9.9% of all the EC populations in the scRNA-seq dataset, in addition to Pre-HE and HE/EHT. Could they please provide these data in supplementary information? How do they define HE (this has never been introduced in the manuscript) on the original UMAP of the scRNAseq (Fig1A)?

Reviewer #2 (Remarks to the Author):

The authors adequately addressed the reviewers' comments.

Reviewer #3 (Remarks to the Author):

The Authors have addressed most of the reviewers concerns.

REVIEWERS' COMMENTS

Reviewer #1 (Remarks to the Author):

“The authors have addressed most of my concerns and comments. I have a few remaining questions that need to be answered.

1. When introducing the CITE-seq experiment, it would be informative to include directly in the text (first sentence of results) the antibodies used on CD31+ cells and the rationale for this choice. I understand they are CD44, CD117, CD41, CD43, CD45.2, and CD27. “

We thank this reviewer for this suggestion and added this information on page 3, lines 7-8 to make it clearer to the readers.

“2. Avoid/eliminate the use of primitive in the sentence “These mature hematopoietic cells, including EMPs, are likely derived from primitive hematopoiesis in the yolk sac but may also represent a few mature cells generated at this stage from the EHT process” as primitive is often used to describe the first wave of primitive erythroid cells that are not giving rise to EMPs.”

We thank this reviewer for catching this mistake. We have now replaced the word primitive with early.

“3. The observation that many endothelial cells express Runx1+23GFP in the absence of Meis1 (Fig6D) is intriguing and interesting. These results suggest that Meis1 is required for EHT instead of preHE specification as proposed (as no RUNX1 expression would be expected in this later case). Could the authors comment on this observation? Could the authors look for Runx1, Gfi1 expression in their scRNA-seq of the Meis1 KO (Fig5D)?”

We agree with this Reviewer that the observation of Runx1+23GFP cells in the absence of Meis1 is interesting. As we discussed in the manuscript, we believe that Meis1 is important for the formation of preHE but not necessarily for the subsequent upregulation of Runx1 through the preHE to HE bottleneck transition previously described [1]. This is consistent with the observation of an EHT cluster in the scRNA-seq in the Meis1-cKO samples and a ratio of EHT to preHE cells that is not significantly different from WT (Fig. 5F). Interestingly, Meis1-cKO EHT cells have a lower EHT gene signature overall compared to WT (Fig. 5H) suggesting that these cells are still missing some important cues. In other words, Meis1 appears to be required to form fully functional preHE cells, and even with Runx1 expression, cells fail to progress through EHT. This could also explain why Runx1+ cells in Fig. 6D (presumably the Runx1-expressing EHT cells in the scRNA-seq) remain embedded in the vascular wall and Meis1-cKO HE failed to produce hematopoietic colonies *ex vivo* (Fig. 5B). It could also be that Meis1 is required at multiple stages (ie: preHE, EHT, and in HSC as has been shown).

We have looked at *Runx1* and *Gfi1* expression in our scRNA-seq dataset. Firstly, as mentioned previously, we do not detect their expression in the preHE cluster. At the EHT stage, we do not see a significant difference in either *Runx1* or *Gfi1* expression at the RNA level in the Meis1-cKO cells compared to WT (see below) and they do not come up in differential gene expression analysis.

"4. It is still incorrect to show in Fig4E some images that do not correspond directly to the amplification of the boxed area, as this would be the expectation of the reader. The data should be presented differently to avoid this ambiguity."

We have replaced the confusing panel in Fig4E with the appropriate image matching the area of interest.

"5. The authors addressed my comment "The RUNX1 immunostaining presented in figure 4E looks suboptimal. More convincing immunostaining should be provided. RUNX1 should be detected in more cells, notably in the sub-aortic mesenchyme." by providing in the revision, new Runx1 staining in figure S6A. However, I was not able to understand this figure S6A in the absence of an annotation or appropriate figure legend. Also, it needs to be clarified how providing a RUNX1 staining alone will address the potential problem with the Meis1/RUNX1/CD31 staining presented in Fig 4E. The staining in Fig 4E should be repeated."

The panel in Suppl Fig 6A shows that Runx1 immunostaining with the antibody used in our experiments labels the same cells as GFP in the Runx1+23GFP reporter mouse model. This goes to support that the antibody is working well under the conditions used in our experiments. We clarified the figure legend for this panel.

During the last revision, we provided additional repeats of Fig. 4E that can be found in Suppl Fig 6B. These immunostaining images show Meis1/Runx1/CD31 co-staining, as requested by this reviewer. We annotated these panels better and clarified the legend in this new version. For Suppl Fig 6B, we can also note the expression of Runx1 in the sub-aortic mesenchyme (also included here below), to address the reviewer's point.

Single panel from supplementary figure 6B (Runx1 in red, CD31 in purple, DAPI in gray, white arrows point to regions with mesenchymal Runx1 detection):

In addition, we are now providing extra images below. This immunostaining shows clearly that we can detect Runx1 (red staining in the left panel) in the sub-aortic mesenchyme (endothelial is marked by CD31 in purple, white arrows point at subaortic Runx1 expression, yellow arrows point at Runx1+ EC). These sub-aortic Runx1+ cells (red) are also Meis1-GFP+ (green in the right panel). We hope that these new images will convince the reviewer that the Runx1 immunostaining works in these experiments.

“6. The authors addressed my concern about a significant discrepancy between the frequency of Meis1 positive ECs detected through CITE-seq (a few %, fig 1A) and the GFP reporter (around 20-30%, fig 3C) by indicating the presence of non-zero counts for Meis1 in 9.9% of all the EC populations in the scRNA-seq dataset, in addition to Pre-HE and HE/EHT. Could they please provide these data in supplementary information? How do they define HE (this has never been introduced in the manuscript) on the original UMAP of the scRNAseq (Fig1A)?”

We added this information in the legend of figure 3E. HE cells were defined in the manuscript on page 3, lines 28-29 (continuing on page 4, line 1-2) as follows: “Phenotypic HE cells expressing CD31⁺CD117⁻CD41⁻CD43⁻CD45⁻ and both *Runx1* and *Gfi1* were located at the extremity of the EHT cluster closest to pre-HE cells in UMAP space (Fig. 1 F, Supplementary Fig. 11). Of note, the presence of these HE cells represented a small fraction of the EHT cluster (13 out of 140 EHT cells)”.

Reviewer #2 (Remarks to the Author):

“The authors adequately addressed the reviewers' comments.”

Reviewer #3 (Remarks to the Author):

“The Authors have addressed most of the reviewers concerns.”

References:

1. Zhu, Q., et al., *Developmental trajectory of prehematopoietic stem cell formation from endothelium*. Blood, 2020. **136**(7): p. 845-856.